



# Seismic imaging across fault systems in the Abitibi greenstone belt – An analysis of pre- and post-stack migration approaches in the Chibougamau area, Quebec, Canada

Saeid Cheraghi[1], Alireza Malehmir[2], Mostafa Naghizadeh[1], David Snyder[1], Lucie Mathieu[3], Pierre Bedeaux[3]

[1] Mineral Exploration Research Centre, Harquail School of Earth Sciences, Goodman School of Mines, Laurentian University, Sudbury, Ontario
[2] Department of Earth Science, Uppsala University, Uppsala, Sweden
[3] Centre d'études sur les ressources minérales (CERM), Département des Sciences appliquées, Université du Québec à Chicoutimi (UQAC), Chicoutimi, Québec

*Correspondence to*: Saeid Cheraghi (scheraghi@laurentian.ca)

**Abstract.**   Two high-resolution seismic reflection profiles acquired north and south of Chibougamau, located in the northeast
of the Abitibi subprovince of Canada help understand historic volcanic-hosted massive sulfide (VMS) deposits and hydrothermal Cu-Au mineralization found there. Major faults crossed by the profiles include the Barlow fault in the north and the Doda fault and the Guercheville fault in the south, all targets of this study that seeks to determine spatial relationships with known metal endowment in the area. Common-offset DMO corrections and common-offset pre-stack time migrations (PSTM) were considered. Irregularities of the trace midpoint distribution resulting from the crooked geometry of both profiles and their
relative contribution to DMO and PSTM method and seismic illumination were assessed in the context of the complex subsurface architecture of the area. To scrutinize this contribution, seismic images were generated for offset ranges of 0-9 km using increments of 3 km. Migration of out-of-plane reflections used cross-dip element analysis to accurately estimate the fault dip. The seismic imaging shows the thickening of the upper crustal rocks near the fault zones along both profiles. In the north seismic reflection section, key geological structure identified include the Barlow fault and two diffraction sets imaged within
the fault zone that represent potential targets for future exploration. The south seismic reflection section shows rather a complicated geometry of two fault systems. The Guercheville fault observed as a subhorizontal reflector connects to a steeply dipping reflector. The Doda fault dips subvertical in the shallow crust but as a steeply dipping reflection set at depth. Nearby gold showings suggest that these faults may help channel and concentrate mineralizing fluids.



## 1 Introduction

Acquiring and image processing of a high-resolution seismic data set over Archean greenstone belts comprised of crystalline rocks characterized by steeply dipping reflectors, point scatters, and multiply folded/faulted structures challenges basic assumptions of the technique (Adam et al., 2000, 2003). During the past 30 years, pre-stack normal moveout (NMO) and dip moveout (DMO) corrections followed by post-stack migration represented the conventional method used in most crystalline-rock case studies globally, with different success rates for both 2D and 3D datasets (Malehmir et al., 2012 and references

therein). The post-stack migration method has provided sharp images in many case studies (Juhlin 1995a; Juhlin et al., 1995; Bellefleur et al., 1998; Perron and Calvert, 1998; Juhlin et al.,2010; Bellefleure et al., 2015; Ahmadi et al., 2013;), however all these studies indicate low signal-to-noise (S/N) ratio and scattering rather than coherent reflection of the seismic waves. Petrophysical measurements, where available, complement reflectivity/velocity models of the shallow crust, i.e., < 1000 m, and permit more accurate correlation of reflections to geological structures (Perron et al., 1997; Malehmir and Bellefleur,

2010). The Kirchhoff pre-stack time/depth migration (PSTM/PSDM) method has also been utilized in crystalline rock environments (e.g. Malehmir et al., 2011; Singh et al., 2019). However, computational complexity and the requirement of a detailed velocity model limited application of a PSTM algorithm (Fowler, 1997). In addition, strong scattering of seismic waves, low S/N ratios, and small-scale changes in acoustic impedance within crystalline rock environments rendered both PSTM and PSDM algorithms less popular in a crystalline rock environment (Salisbury et al., 2003; Heinonen et al., 2019;

Singh et al., 2019; Braunig et al., 2020). An important, somewhat neglected issue is the effect of survey geometry on processing results and if it is possible to adjust the processing flow to compensate. An optimized processing flow appears essential to image deep mineral deposits and structures such as faults which host base/precious metal deposits (Malehmir et al., 2012 and references therein).

Apart from the type of migration method (i.e., post-stack migration, PSTM or PSDM), the survey design parameters such as survey length, orientation, number of shots and receivers, shot and receiver spacing are major factors that affect the seismic illumination for both 2D and 3D surveys (Vermeer, 1998). A seismic study in Brunswick, Canada, showed that 2D seismic surveys provided high-resolution seismic images of the upper crust but a 3D survey acquired over the same area failed to provide more details mostly because of survey design (Cheraghi et al., 2011 and 2012). Typically, crystalline rock seismic

surveys in forested countries use crooked line profiling along forest tracks or roads for logistic and ultimately economic or environmental considerations. While 2D seismic processing algorithms are designed to work on straight survey lines with regular offset distribution of trace midpoint (CMPs), the crooked surveys violate those assumptions and need compensating strategies such as dividing the crooked survey into several straight lines, 3D swath processing, or cross-dip analysis (Adam et al., 1998; Milkereit and Eaton, 1998; Adam et al., 2000; Schmelzbach et al., 2007; Kashubin and Juhlin, 2010). Beside the

processing strategy, the offset distribution affects seismic illumination wherever an essential processing step such as common-offset DMO corrections or common-offset Kirchhoff PSTM algorithm is applied (Fowler, 1997 and 1998). Proficiency of both



these methods demands a regular distribution of source-receiver offsets because of their sensitivity to constructive contribution of offset planes (Canning and Gardner, 1998; Cheraghi et al., 2012; Bellefleur et al., 2018; Braunig et al., 2020).

This case study focuses on seismic sections along two 2D high-resolution profiles, the south and north surveys, respectively (Fig. 1), acquired in 2017 in the Chibougamau area, Quebec, Canada. These profiles were acquired to aid upper crustal-scale studies of metal-endowed fault structures. The Chibougamau area mostly hosts VMS (e.g., Mercier-Langevin, et al., 2014) and Cu-Au magmatic-hydrothermal mineralization (Pilote et al., 1997; Mathieu and Racicot, 2019). Orogenic Au mineralization also documented in this area (Leclerc et al., 2017) typically relates spatially to crustal-scale faults; hence, the

importance to document the geometry of major fault during exploration (Groves et al., 1998; Phillips and Powell, 2010). In order to image fault systems in the Chibougamau area, we generated DMO stacked migrated sections as well as images generated with a PSTM algorithm. We inclusively investigated the surveys' acquisition geometries and their effects on the DMO and PSTM to optimize these processing flows according to the specific geometry. We compare the results from both methods to show the strategy and criteria used to design attributes of our processing flow that favor the acquisition geometries

of each profile to enhance coherency of the seismic reflections in both shallow and deeper crust. To accomplish this goal, we: (1) apply pre-stack DMO corrections followed by post-stack migration along both profiles; (2) analyze the application of a PSTM algorithm on both surveys; (3) specifically test the CMP offset distribution and its contribution to DMO corrections and PSTM with an offset range of 0-10 km; and (4) address the effect of cross-dip offsets and their relevant time shifts on the imaged reflections. Our optimized application of DMO and PSTM contributes information on the geometry of the faults in the

Chibougamau area, which is essential to understand mineralization potential in the area and to target regions of higher prospectivity.

**2 Geological setting**

The Chibougamau area is located in the northeast portion of the Neoarchean Abitibi subprovince (Fig. 1). The oldest rocks in the study area (> 2760 Ma; David et al., 2011) include mafic and felsic lava flows as well as volcanoclastic deposits of the

Chrissie and Des Vents formations (Fig. 1, see Leclerc et al., 2017; Mathieu et al., 2020b). These rocks are overlain by sedimentary and volcanic rocks of the Roy Group, emplaced between 2730 and 2710 Ma and which constitute most of the covered bedrock (Leclerc et al., 2017; Mathieu et al., 2020b). The Roy Group includes a thick (2-4 km) pile of mafic and intermediate volcanic rocks topped by a thinner assemblage of lava flows, pyroclastic and sedimentary units (volcanic cycle 1, Leclerc et al., 2012 and 2015), as well as a pile of mafic lava flows capped by a thick (2-3 km in the north to 0.5 km in

south) succession of intermediate to felsic flows and fragmental units interbedded with sedimentary rocks (volcanic cycle 2). The Roy Group is overlain by sandstone and conglomerate of the 2700-2690 Ma Opémisca Group, which accumulated in two sedimentary basins (Mueller et al. 1989, Leclerc et al. 2017). The main rock exposures of the Roy Group observed along the



southern profile consist of pelitic to siliciclastic sedimentary rocks of the basin-restricted Caopatina Formation (volcanic cycle 1 or Opémisca Group) and mafic to intermediate lava flows of the Obatogamau Formation (volcanic cycle 1).

**Figure 1:** The geological map over the Chibougamau study area which major fault zones are marked. The regional seismic survey and the high-resolution seismic surveys in north and south of the area are presented and some of the CDP locations are marked. The inset shows the location of the study area.





The rock units around the north profile include the Bruneau Formation (mafic lava flows), the Blondeau Formation (intermediate to felsic, volcanic, volcanoclastic and sedimentary deposits), and the Bordeleau Formation (volcanoclastic deposits, arenite, conglomerate) of volcanic cycle 2, as well as sedimentary rocks of the Opémisca Group (Dinmroth et al., 1995; Leclerc et al., 2012). The major intrusions relevant in the study area are the ultramafic to mafic sills of the Cummings
Complex, which intrude the lower part of the Blondeau Formation (Bédard et al., 2009).

Several east-trending fault zones and synclinal/anticlinal systems are associated to Neoarchean deformation events in the Chibougamau area (Dimroth et al., 1986; Daigneault and Allard 1990; Daigneault et al., 1990; Leclerc et al., 2012; Leclerc et al., 2017). The main faults, folds and associated shistosity and metamorphism relate to a Neoarchean N-S shortening event
(Mathieu et al., 2020b and references therein). The north survey lies nearly perpendicular to the major regional structures. It crosses the west-striking Barlow fault zone, a shallowly to steeply south-dipping fault zone (Sawyer and Ben, 1993; Bedeaux et al., 2020). The north survey also crosses the Waconichi syncline and the steeply dipping, east to west striking faults of the Waconichi Tectonic Zone (Fig. 1). The south survey passes through the Guercheville fault zone, which intersects the Druillettes syncline (Fig. 1), and north of the east-striking Doda fault zone. The Doda fault zone appears subvertical at surface (Daigneault,
1996); the Guercheville fault dips northward at 30-60 degrees but was mapped locally as a subvertical fault (Daigneault, 1996). Most of these faults form early basin-bounding faults (Opémisca basins) reactivated during the main shortening event (Dimroth, 1985, Mueller et al. 1989).

## 3 Seismic data acquisition

The 2017 seismic survey in the Chibougamau area forms part of the Metal Earth exploration project in the Abitibi greenstone
belt (Naghizadeh et al., 2019). High-resolution seismic segments in the north and south augment a coincident regional seismic line that crosses the main geological structures of the area (Fig. 1). Cheraghi et al. (2018) demonstrated that the Chibougamau regional survey capably imaged reflections in both the upper and lower crust (down to Moho depth). Mathieu et al. (2020b) interpreted the regional seismic survey to map major faults and structures in relation to potential metal endowment.

The high-resolution surveys in the Chibougamau area form the focus of this study. In total, the survey acquired 2281 vibrator points (VPs) along the north survey and 3126 VPs along the south survey (Fig. 1). Consistent with other high-resolution surveys in the Metal Earth project (Naghizadeh et al., 2019), shot and receiver spacing were set at 6.25 m and 12.5 m, respectively, with a sampling rate of 2 ms. Detailed attributes of both surveys are shown in Table 1.




**Table 1: Data acquisition summary of the high-resolution Chibougamau north and south surveys (year 2017)**

| | High-resolution survey (R2) |
|---|---|
| Spread type | Split spread |
| Recording instrument | Geospace GSX Node |
| Field data format | SEGD (correlated) |
| Geophone type | 5 Hz, single component |
| Source type | VIBROSEIS |
| No. of sources | 3 |
| Sweep length (s) | 28 |
| No. of Sweeps | 1 |
| Source starting frequency (Hz) | 2 |
| Source ending frequency (Hz) | 120 |
| Field low cut recording filter (Hz) | 2 |
| Field high cut recording filter (Hz) | 207 |
| Record length (s) | 12 after cross-correlation |
| Sampling rate (ms) | 2 |
| Shot spacing (m) | 6.25 |
| Receiver spacing (m) | 12.5 |
| Nominal maximum offset for processing (km) | 10 |
| Number of acquired shots | 2281[a] & 3126[b] |
| Survey length (km) | ~15[a] & ~19[b] |

[a] North survey   [b] South survey

### 3.1 Offset distribution for Kirchhoff PSTM and DMO corrections

Based on the analysis shown in Appendix A, both profiles could record alias-free P-wave energy at velocities necessary for seismic imaging in crystalline rock environments, i.e., greater than 5000 ms$^{-1}$. Our analysis also indicates that both profiles are alias-free for shear waves and low velocity noise, e.g., ground roll. We investigated the Chibougamau profiles to evaluate irregularity and optimize the application of PSTM and DMO corrections. The offset distribution forms our main criteria with which to investigate the relative quality of pre- and post-stacked migrated images in the Chibougamau area based on common-offset PSTM (Fowler, 1997) and common-offset DMO correction (Hale, 1991; Fowler, 1998). In Appendix A we show the necessity of regular offset distribution when using common-offset DMO or PSTM (Fig. A1). Other methods of DMO or PSTM such as Common-azimuth PSTM (Fowler, 1997) and common-azimuth DMO corrections should theoretically provide results equal to those assuming common-offset (Fowler, 1997 and 1998). Our study did not analyze common-azimuth algorithms. Besides the effect of regularity/irregularity of the survey, we also explain in Appendix A that not necessarily all CMPs contribute to the DMO process (DMO illumination concept). Optimized DMO illumination can be investigated during survey design by testing different subsurface models or survey geometries (Beasley, 1993). The common-offset DMO and common-offset PSTM utilize similar algorithms for migration (Fowler, 1997 and 1998) and the illumination concept applies to PSTM as well.






The maximum offset in these Chibougamau surveys is 10 km. We evaluated if specific offset values contribute constructively or destructively in the resulting PSTM or whether they generate artefacts during the DMO corrections. We also investigated PSTM and DMO corrected images at different offsets to find the offset range that optimizes subsurface illumination (Vermeer, 1998).


For the Chibougamau profiles, we evaluated CMP distributions within CDP bins (6.25 m, Table 2) along each survey. Figs. 2 and 3 present examples of CMP offset/azimuth distribution along the north and south surveys, respectively. Some of the CDP bins show a regular offset distribution, for example Fig. 2b and 2c from the north profile or Fig. 3b from the south profile, respectively; note that bins located in the middle of the survey have short and long offsets equally mapped north and the south of the bin center). The azimuth distribution of these CDP bins also shows a symmetric pattern relative to the CDP line directions, for example Fig. 2f and 2g from the north profile and Fig. 3e from the south profile. Some of the CDP bins however, present irregular offset and asymmetric azimuth distributions, for example Fig. 2a, 2d, 2e, and 2h from the north profile, and Fig. 3c and 3f from the south profile, respectively. These CDP bins show that longer offsets are mapped unevenly in the bins resulting in an asymmetric azimuth distribution pattern. The analysis indicates that most of the irregularity of offset distribution occurs due to a lack of longer offsets in those bins.

Based on the analysis shown in Figs. 2 and 3 and evaluating the distribution pattern of offset for the north and south profiles, we predict irregular distribution of CMPs would be a challenge for 2D PSTM and DMO corrections. Another challenge is whether CMPs of profiles acquired in the Chibougamau area contribute constructively in DMO/PSTM towards subsurface illumination considering the geometry of specific reflectors, i.e., dip and strike (more details in Appendix A). We designed offset planes ranging 0-3 km, 0-6 km, and 0-9 km in order to study the survey geometry (Fig. 4). In the north profile, CMPs with offsets ≤ 6 km cluster along the survey line (Fig. 4a and 4b) whereas many CMPs with offsets greater than 6 km do not (Fig. 4c). The CMPs of the south profile lie along the survey line for all offset ranges (Fig. 4d, 4e, and 4f) due to the less crooked pattern of the south profile compared to than the north profile (Fig. 4).



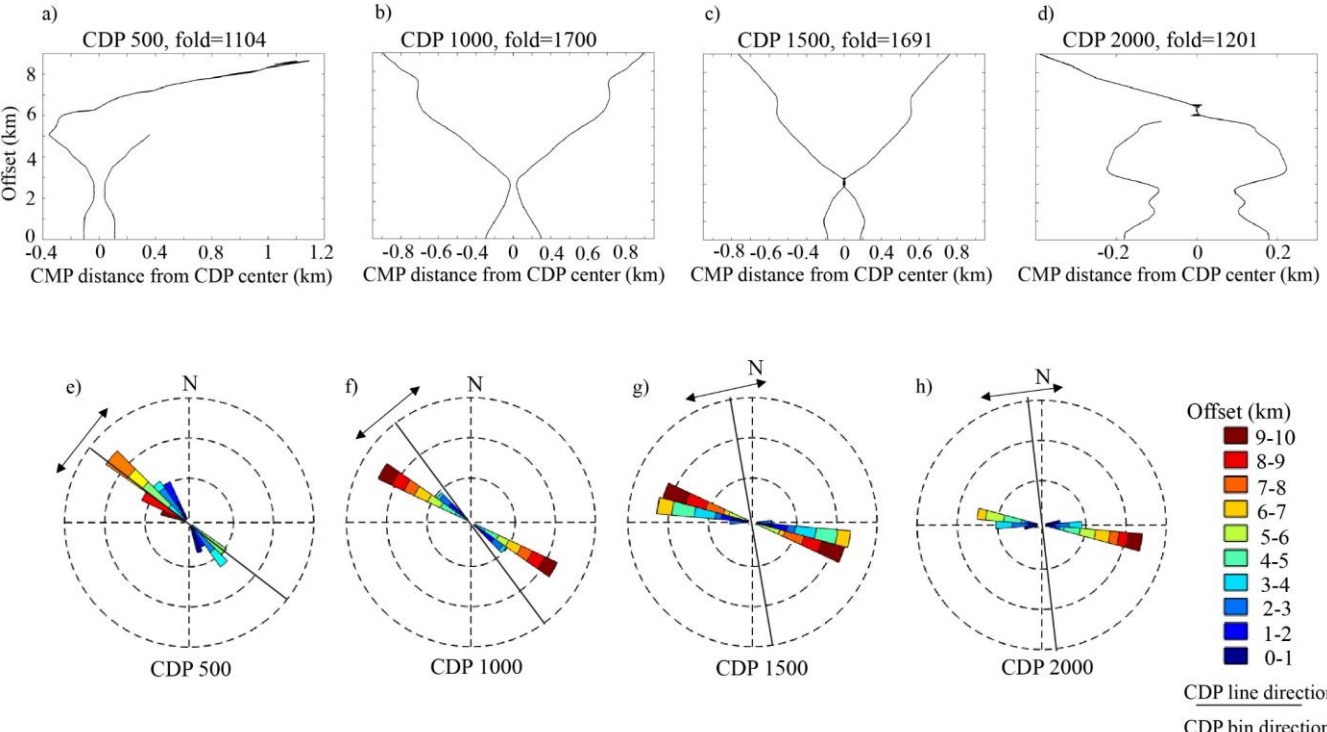

**Figure 2:** CMP offset and azimuth distribution from the north survey. The offset distribution is shown for (a) CDP 500, (b) CDP 1000, (c) CDP 1500, and (d) CDP 2000. See Figs. 1 and 4 for the location of the CDPs. The negative values for CMP distance in graphs (a)-(d) indicate CMP is located in the south of the bin center and the positive values implies that CMP is located in the north of the bin center. The azimuth distribution is shown for (e) CDP 500, (f) CDP 1000, (g) CDP 1500, and (h) CDP 2000. For each diagram shown in (e)-(h) the CDP line direction is presented. The CDP bin is perpendicular to the CDP line.





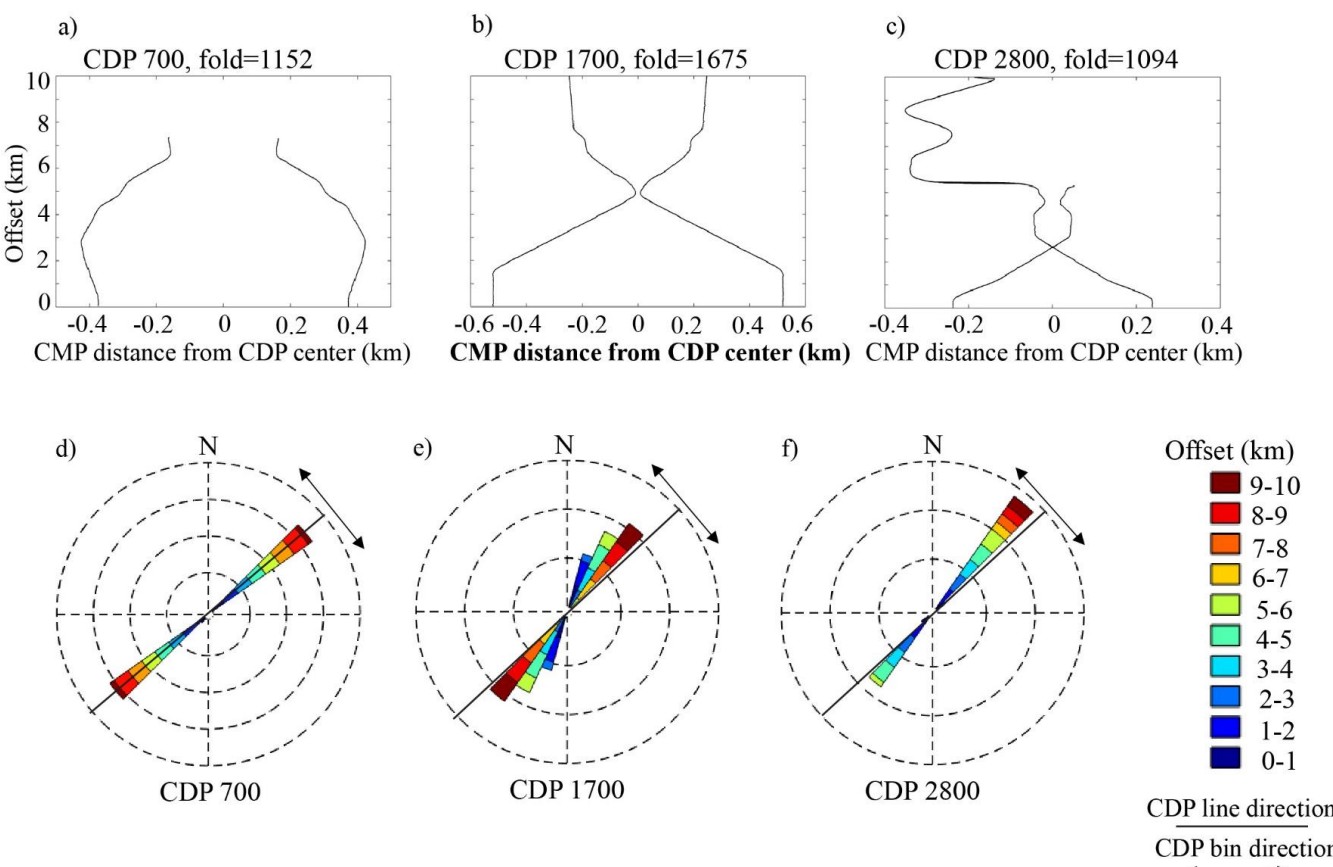

**Figure 3:** CMP offset and azimuth distribution from the south survey. The offset distribution is shown for (a) CDP 700, (b) CDP 1700, and (c) CDP 2800. See Figs. 1 and 4 for the location of the CDPs. The negative values for CMP distance in graphs (a)-(c) indicate CMP is located in the south of the bin center and the positive values implies that CMP is located in the north of the bin center. The azimuth distribution is shown for (d) CDP 700, (e) CDP 1700, and (f) CDP 2800. For each diagram shown in (d)-(f) the CDP line direction is presented. The CDP bin is perpendicular to the CDP line.



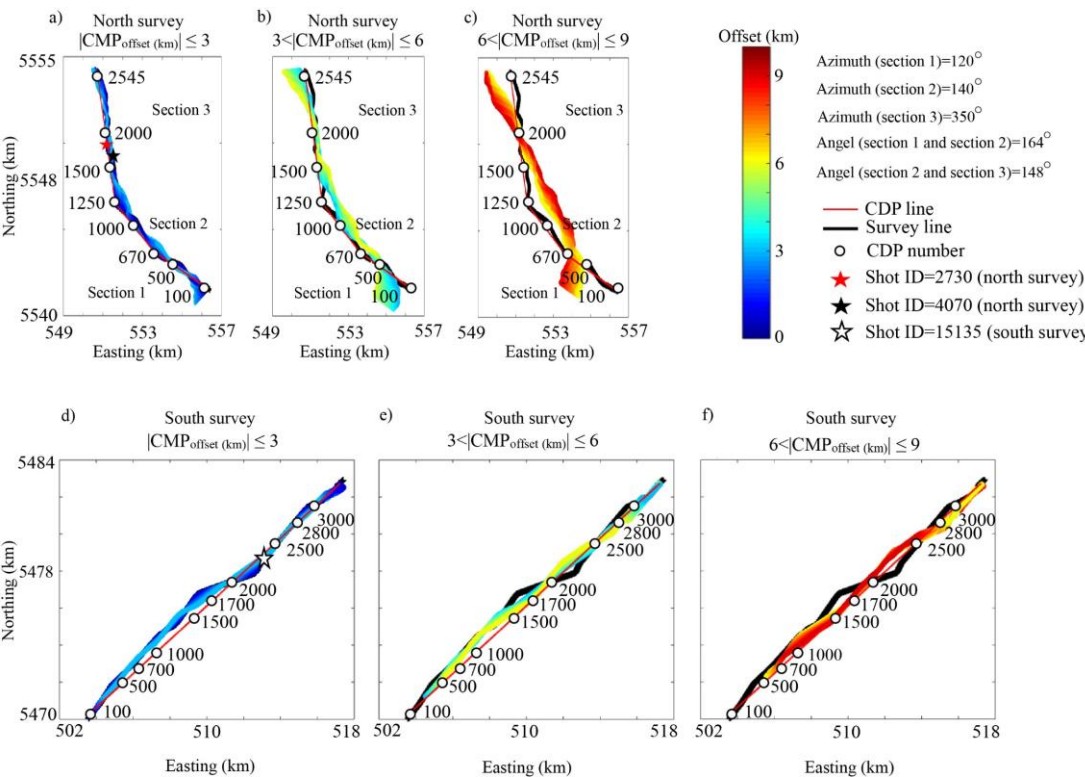

**Figure 4:** CMP offset distribution at range of 0-10 km for the north and the south survey in Chibougamau area. The distribution for the north survey is shown for (a) $\left|CMP_{offset(km)}\right| \leq 3$, (b) $3 < \left|CMP_{offset(km)}\right| \leq 6$, and (c) $6 < \left|offsetCMP_{offset(km)}\right| \leq 9$. The distribution for the south survey is shown for (d) $\left|CMP_{offset(km)}\right| \leq 3$, (e) $3 < \left|CMP_{offset(km)}\right| \leq 6$, and (f) $6 < \left|offsetCMP_{offset(km)}\right| \leq 9$. The CDP line and the survey line is shown in the figure. Some shot and CDP location is also shown. The azimuth of each section of the CDP line from the north survey and the angel between two sequential sections is presented.



**Table 2: Processing parameters and attributes for the Chibougamau surveys**

| | Chibougamau north and south surveys |
|---|---|
| 1 | Read data in SEGD format and convert to SEGY for processing |
| 2 | Setup geometry, CDP spacing of 6.25 m |
| 3 | Trace editing (manual) |
| 4 | First arrival picking and top muting (0-10000 m offset) |
| 5 | Elevation and refraction static corrections (replacement velocity 5200 ms$^{-1}$, V0 1000 ms$^{-1}$) |
| 6 | Spherical divergence compensation (velocity power of 2 and travel time power of 1, V$^2$t ) |
| 7 | Median velocity filter (1400, 2500, 3000 ms$^{-1}$) |
| 8 | Band-pass filter (5-20-90-110  Hz) [a & b] |
| 9 | Airwave filter |
| 10 | Surface-consistent deconvolution [c & d] |
| 11 | Trace balancing |
| 12 | AGC (window of 150 ms) |
| 13 | Velocity analysis (iterative) |
| 14 | Surface consistent residual static corrections |
| 15 | DMO corrections [a & b] (5500 ms$^{-1}$, offset range of 0-3 km, 0-6 km, and 0-9 km) |
| 16 | Velocity analysis (iterative at range of 5000-6500 ms$^{-1}$) |
| 17 | Stacking |
| 18 | Coherency filter [e & f] |
| 19 | Trace balancing |
| 20 | Phase-shift time migration (north survey: 5000 ms$^{-1}$; south survey: 5300 ms$^{-1}$) |
| 21 | Kirchhoff PSTM [a & b] (after step 14 shown in here; offset range of 0-3 km, 0-6 km, and 0-9 km) |
| 22 | Time to depth conversion (6000 ms$^{-1}$ for both north and south surveys) |

[a & b] This is applied to both north and south surveys.

[c] North survey the filter length and gap is 100 ms and 16 ms, respectively

[d] South survey: the filter length and gap is 100 ms and 18 ms, respectively

[e] North survey: F-X deconvolution, filter length of 39 traces

[f] South survey: F-X deconvolution, filter length of 19 traces

## 4 Data processing and results

We considered a pre- and post-stack processing workflow for both the north and south profiles similar to that applied by
240 Schmelzbach et al. (2007), and generated migrated DMO-corrected stacked sections as well as Kirchhoff PSTM sections
(Table 2). The CMP distribution of the Chibougamau south survey lies mostly along a straight line hence a linear CDP
processing line was designed (Fig. 4). The CMP coverage along the north profile follows a crooked pattern hence a curved
CDP line that smoothly follows this geometry was used (Fig. 4). The main processing steps included attenuation of
coherent/random noise, refraction and residual static corrections, sharpening the seismic data using a deconvolution filter and
245 a top-mute to remove first arrivals.



Based on the aforementioned analysis, we considered offset ranges of 0-3 km, 0-6 km, and 0-9 km, for DMO corrections and the PSTM. These steps were also deemed necessary:

- Reflection residual static corrections were applied to all shot gathers prior to the DMO corrections and PSTM application (steps 1-14 in Table 2).

- Constant DMO corrections with a velocity of 5500 ms$^{-1}$ were applied for both the north and south surveys. This chosen velocity derived from several tests using various constant velocities, 5000-6500 ms$^{-1}$ with step range of 100 ms$^{-1}$.

- After DMO corrections, velocity analysis with constant stacking velocity in the range of 5000-6500 ms$^{-1}$ helped to design an optimized velocity model for NMO corrections and the stacking (Table 2).

- Choosing a velocity model for PSTM was a time consuming procedure performed on the basis of trial and error. We tried constant velocity models at a range of 5000-6500 ms$^{-1}$ (step rate of 100 ms$^{-1}$) as well as the velocity model applied for the DMO-NMO correction (see above). The best model adopted velocities within 90-110 % of the DMO velocity model.

The DMO corrected migrated stacked sections and PSTM sections of the north and south survey appear in Figs. 5 and 6, respectively. The offset range of 0-3 km reveals the most coherent reflections for both methods (Figs. 5a-b and 6a-b); the velocity analysis after DMO corrections significantly improved the coherency of the reflections for the sections with an offset range of 0-3 km (Figs. 5a and 6a). The migrated sections generated from offset ranges of 0-6 km and 0-9 km (Figs. 5c-f, and 6c-f) failed to improve the stacked sections. The best results of the stacked sections from the longer offsets (Figs. 5c, 5e, and 6c, 6e) were observed with a velocity model similar to the one applied to Figs. 5a and 6a for stacking after DMO correction.

The design of the north survey CDP line used three segments: CDPs 100-670 have an azimuth of 120°, CDPs 670-1250 have an azimuth of 140°, CDPs 1250-2545 have an azimuth of 350° (Fig. 4). Table 3 indicates geometrical attributes of key reflections imaged along the north profile. The first segment, ending at the contact between sedimentary rocks of the Bordeleau Formation and mafic rocks of the Bruneau Formation, appears seismically transparent without any prominent reflections (Fig. 5a and 5b). Labelled as in Fig. 5, chn1, chn2, and chn3 mark the major reflections imaged in the upper crust. The most prominent reflection package of the north survey is chn3, with an apparent width of approximately 3 km on the surface and an apparent thickness of approximately 2 km (see Table 3 for detailed attributes). Reflections chn4, chn5, and chn6 image at depths greater than 2 km and do not show any correlation to the surface geology. The horizontal reflection chn_diff, with a horizontal length of approximately one kilometer, appears in the DMO staked migrated section (Fig. 5a) and also weakly in the PSTM section (Fig. 5b). Reflection chn_diff intersects the chn4 reflections. The apparent geometry of the chn_diff reflection in the migrated sections would suggest a curved feature or else a diffracted wave that collapsed to a horizontal reflection after the migration.

The Chibougamau south survey mostly traverses mafic to intermediate lava flows of the Obatogamau Formation and sedimentary rocks of the Caopatina Formation (Fig. 6). The DMO stacked migrated (Fig. 6a) and PSTM sections (Fig. 6b) both show steeply dipping and subhorizontal reflections in the upper crust, but upper crustal reflections in the DMO stack section (Fig. 6a) show more coherency than those of the PSTM (Fig. 6b). Therefore, the DMO stack facilitates correlation with





the surface geology. Reflection packages chs1, chs2, and chs3 mark the most prominent features in the upper crust imaged along the south survey. The deeper reflections include reflection chs4 at depths greater than 2 km and two packages of subhorizontal reflections chs5 and chs6 at depths greater than 6 km, together extended along 18 km length of the survey. Table 3 summarize the geometrical attributes of these reflections.

## 5 Data analyses

The analysis performed on offset distribution indicated that selecting a proper offset range, here 0-3 km, was crucial for both DMO corrections and PSTM. Another factor that could affect the imaging involves CMP locations relative to CDP bin centers. For the Chibougamau surveys, the maximum CMP offset perpendicular to the CDP line was about ±0.4 km when an offset range of 0-3 km is considered for processing (Fig. 4a and 4d). The 3D nature of subsurface geology around a crooked-line survey requires that out-of-plane features be evaluated, accounting for the time shifts from these features. The out-of-plane

CMPs scatter/reflect seismic waves from steep structures off the CDP line (cross-dip direction); cross-dip analysis addresses time shifts of those structures and adjusts accordingly (for example, Larner et al., 1979; Bellefleur et al., 1995; Nedimovic and West, 2003; Rodriguea-Tablante et al., 2007; Lundberg and Juhlin, 2011; Malehmir et al., 2011). Calculated time delays, called cross-dip move out (CDMO) and treated as static shifts can be applied to both NMO or DMO corrected sections (Malehmir et al., 2011; Ahmadi et al., 2013;). CDMO is sensitive to both velocity and the cross-dip angle applied, however, the variation of

the angle appears more crucial for hard rock data (Nedimovic and West, 2003).

In this Chibougamau case study, we used DMO corrected sections (Table 2) for CDMO analysis, similar to a study by Malehmir et al. (2011). First, the CMP offset relevant to a bin center and perpendicular to the CDP line was calculated (Fig. 4). A constant velocity of 5500 ms⁻¹ was selected for the CDMO analysis. CDMO calculated for dip angles varying from 40°

to west to 40° to east with a step rate of 2° was then applied to DMO corrected CMPs. Finally, we stacked DMO-CDMO corrected traces using a velocity model designed from the one applied after DMO corrections during standard processing (Table 2). Further velocity analysis checked if the coherency of the reflections could be improved, but the new velocity model, where different, showed less than ±5 % changes from the input model. Some example of the CDMO analysis applied to the Chibougamau surveys appears in Figs. 7-9. Table 3 summarizes which CDMO elements (i.e., toward east or west or no cross-

dip) increase the coherency of the reflections when considering time delays associated with out-of-plane reflections.

In the Chibougamau north survey, most of seismic reflectivity is observed at CDPs 700-2500 (Figs. 4 and 5), which include segments 2 and 3 of the processing line; as such, we have performed the CDMO analysis for those two sections, separately. In segment 2 (CDPs 670-1250, Fig. 4), reflections chn1, chn2, and chn3 appear with no cross-dip element applied (Fig. 7c). The

CDMO analysis of segment 2 (Fig. 7) did not reveal any significant reflectivity in deeper part of the section (i.e., 6-12 km, mid-crust). Table 3 shows the optimized CDMO elements for segment 2. The CDMO analysis along segment 3 is shown as



Fig. 8, and Table 3 shows the optimized CDMO results for this segment. The DMO-CDMO stacked sections are essential for the diffraction imaging. Applying the westward CDMO increased the coherency of the diffraction chn_diff. A diffraction package imaged at depths less than 3 km (dashed area in Fig. 8c) is not imaged in the migrated sections (Fig. 5). One horizontal

reflection at a depth of approximately 11 km between CDPs 1600-2000 located within reflection package chn6 shows almost equal coherency independent of the applied cross-dip to east or west (Fig. 8).

The CDMO analysis in the south profile was more challenging because of interfering reflections that dip steeply to the north and to the south (Fig. 6). The CDMO analysis results for the south survey appear in Fig. 9 and Table 3. The reflection chs2

displays a complicated CDMO analysis (Fig. 9). With cross-dip towards the west assumed, reflection chs2 becomes less steep (Fig. 9). Assuming a cross-dip of 30° to west, chs2 dips 20° to the south (Fig. 9a) whereas with no CDMO correction it dips 40° to south (Fig. 9c). With any cross-dip element towards the east applied, chs2 dips more steeply. Reflection chs2 dips 50° to the south with a cross-dip element of 40° to the east applied (Fig. 9f). CDMO analysis for reflection chs3, presents another complicated scenario. This reflection shows the same dip (40°) and its coherency improves with increasing west cross-dip

element (Fig. 9a, 9b, and 9c). On the other hand, with an east cross-dip element applied, reflection chs3 becomes less steep (for example 20° in Fig. 9e versus 40° in Fig. 9c) and its coherency decreases (Fig. 9c-f).






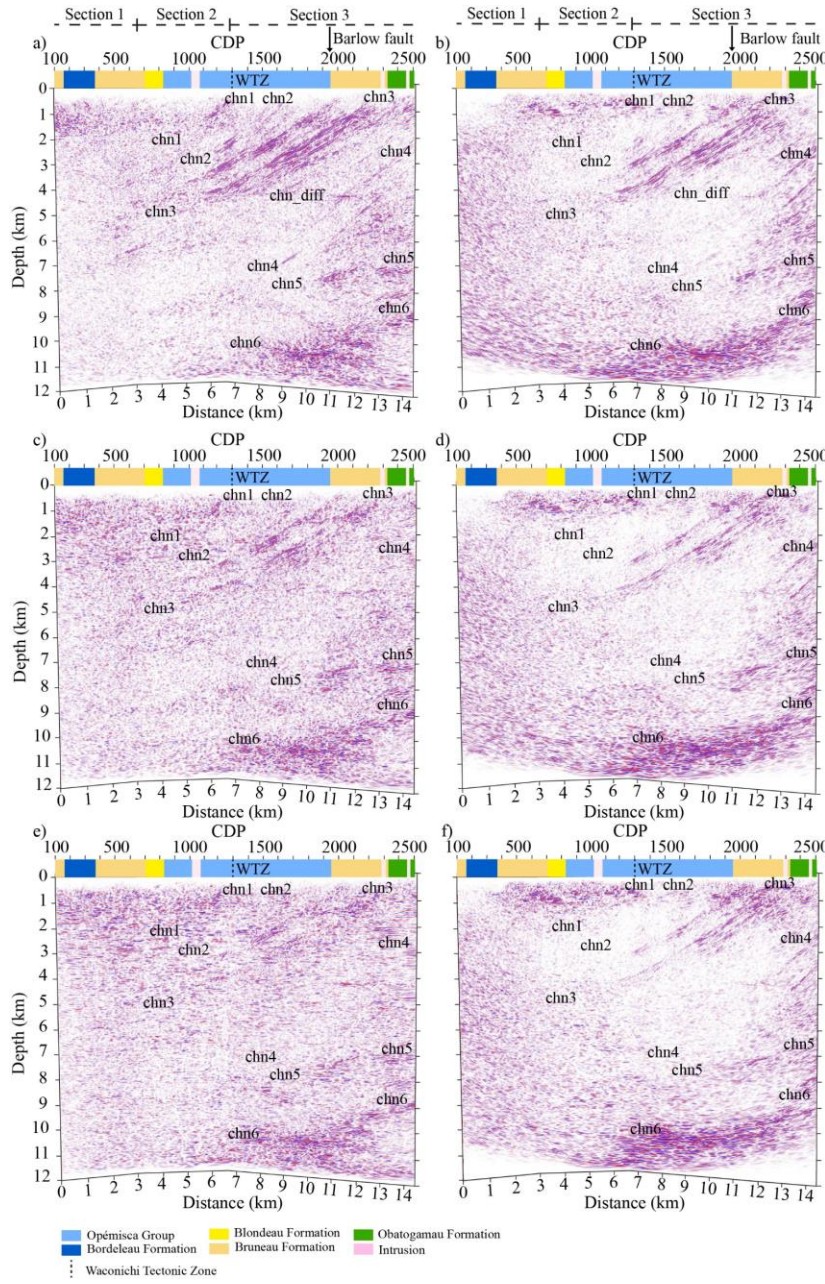

**Figure 5:** Migrated sections from the north survey with considering offset plane at range of 0-10 km. DMO corrected migrated section and PSTM section shown in (a) and (b) for offset plane of 0-3 km, respectively; and shown in (c) and (d) for offset plane of 0-6 km, respectively; and shown in (e) and (f) for offset plane of 0-9 km, respectively. Prominent reflections are imaged in shallow and deep zone of the sections. For interpretation of chn1, chn2, chn3, chn4, chn5, chn6, and chn_diff see text. The survey includes 3 sections which are projected on top





of the image. The rock units along the survey path are projected on top of each section with no dip in the contacts implied. The surface location of the Barlow fault is marked on top of the section.

**Table 3: Geometrical attributes of reflections imaged in Chibougamau area**

| | Reflection name | CDP location | Dip (°) | Dip direction | Subsurface extension | CDMO Segment 2 | CDMO Segment 3 |
|---|---|---|---|---|---|---|---|
| **North profile** | | | | | | | |
| | chn1[PF] | 800-1300 | 40 | South | Near surface down to ~ 2 km | No cross-dip | - |
| | chn2[PF] | 900-1700 | 40 | South | Near surface down to ~ 3 km | 10° to east | 10° to east |
| | chn3[GC,BF] | 1000-2500 | 30 | South | Near surface down to ~ 5 km | 10° to east | 10° to east |
| | chn4[PF] | 1500-2600 | 40 | South | 2-7 km | - | No cross-dip |
| | chn5[GC] | 1800-2600 | Subhorizontal | South | 7-12 km | - | 12° to west |
| | chn6[GC] | 1400-2600 | Subhorizontal | South | 7-12 km | - | 30° to west |
| | chn diff | 1900-2000 | Horizontal | - | At depth of ~ 4 km | - | 12° to west |
| **South profile** | | | | | | **CDMO** | |
| | chs1[GC] | 1600-1700 | 40 | South | Near surface down to ~ 3 km | No cross-dip | |
| | chs2[GC,PF,GV] | 1700-2800 | 40 | South | 1-5 km | Complicated structure for CDMO analysis* | |
| | chs3[GC] | 600-1800 | 40 | North | Near surface down to ~ 7 km | Complicated structure for CDMO analysis* | |
| | chs4[GC,PF,DF] | 100-800 | 30 | North | 2-5 km | 30° to west | |
| | chs5[GC] | 100-1700 | Subhorizontal | North | 6-9 km | 30° to west | |
| | chs6[GC] | 1700-2700 | Subhorizontal | South | 6-9 km | 10° to east | |

*The reflection package shows varying dip with cross-dip to east or west applied. See text for more details.

[GC] The geological contact    [PF] The possible fault    [BF] The Barlow fault    [GV] The Guercheville fault    [DF] The Doda fault

## 6 Discussion

The high-resolution seismic profiles acquired in the Chibougamau area present an essential case study to address challenges of application of the method in crystalline rock environment. One goal of our research was to adjust the processing flow to improve subsurface illumination. To achieve this, we analyzed the performance of common-offset DMO and PSTM. Another aspect of our research involved geologic interpretation of the seismic sections, especially around the fault zones, that could unravel potential zones for detailed mineral exploration.

### 6.1 The effect of survey geometry on seismic imaging

The analysis performed on common-offset DMO and PSTM sections showed the importance of offset range and CMP distribution on CDP bins and whether CMPs offsets at ranges of 0-10 km could all contribute constructively in the resulting images (Figs. 5 and 6). The analysis summarized in Figs. 2 and 3 indicates that the survey geometry resulted in irregular offset distribution in CDP bins, especially for longer offsets. The immediate effect of this irregularity was under-performance of DMO and PSTM for the longer offsets (Figs. 5 and 6). We explain in Appendix A that several factors including spatial attributes of the reflectors (i.e., dip and strike) and survey geometry (i.e., shot and receiver location) define the DMO illumination. Ideally, the impact of known subsurface architecture on DMO illumination should be analyzed before data acquisition at the survey design stage (Beasley, 1993; Ferber, 1997). In our study the DMO illumination criteria can be extended to the PSTM process because common-offset DMO correction and common-offset PSTM utilize similar algorithms for migration (Fowler, 1997 and 1998).







**Figure 6:** Migrated sections from the south survey with considering offset plane at range of 0-10 km. DMO corrected migrated section and PSTM section shown in (a) and (b) for offset plane of 0-3 km, respectively; and shown in (c) and (d) for offset plane of 0-6 km, respectively; and shown in (e) and (f) for offset plane of 0-9 km, respectively. Prominent reflections are imaged in shallow and deep zone of the sections. For interpretation of chs1, chs2, chs3, chs4, chs5, and chs6 see text. The rock units along the survey path are projected on top of each section with no dip in the contacts implied. The surface location of the Guercheville fault is marked on top of the section.





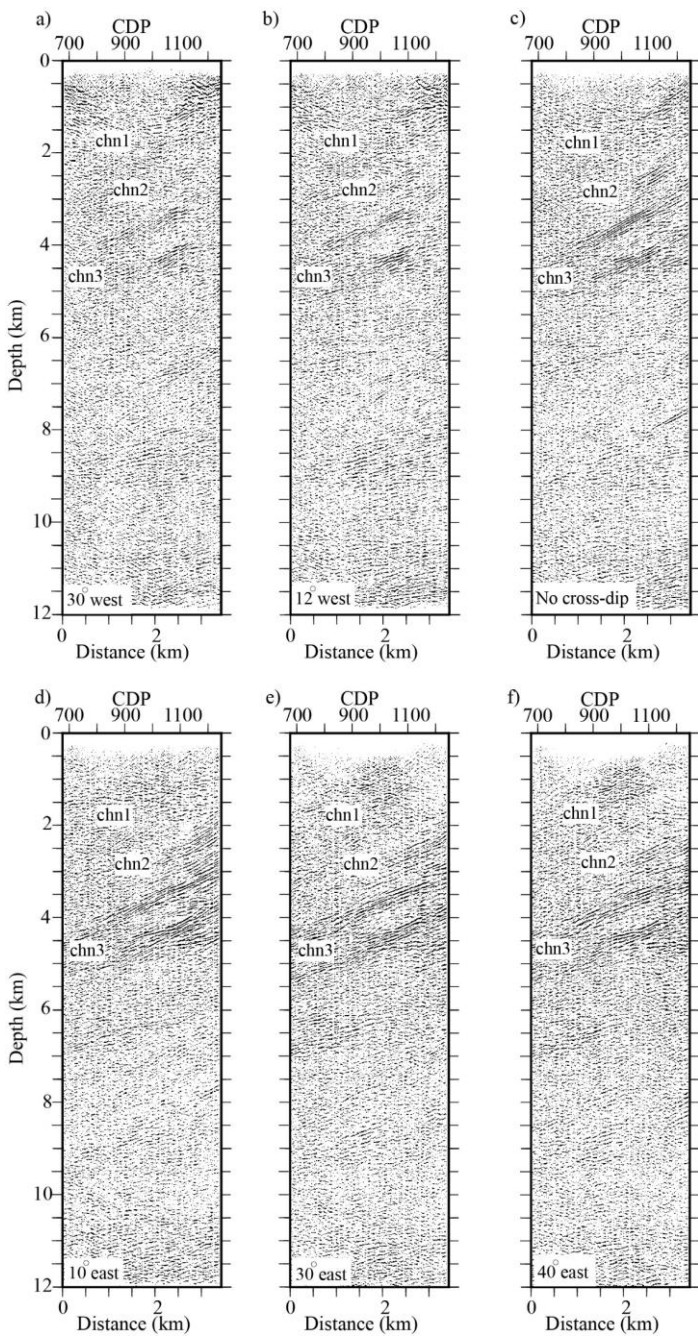

**Figure 7:** CDMO analysis for the north survey along section 2 (see Fig. 4 for the location of the section). (a) DMO corrected stacked section with cross-dip element of 30° to west applied. (b) DMO corrected stacked section with cross-dip element of 12° to west applied. (c) DMO corrected stacked section with no cross-dip element applied. (d) DMO corrected stacked section with cross-dip element of 10° to east applied. (e) DMO corrected stacked section with cross-dip element of 30° to east applied. (f) DMO corrected stacked section with cross-dip element of 40° to east applied. see text for interpretation of marked reflections.





**Figure 8:** CDMO analysis for the north survey along section 3 (see Fig. 4 for the location of the section). (a) DMO corrected stacked section with cross-dip element of 30° to west applied. (b) DMO corrected stacked section with cross-dip element of 12° to west applied. (c) DMO corrected stacked section with no cross-dip element applied. (d) DMO corrected stacked section with cross-dip element of 10° to east applied. (e) DMO corrected stacked section with cross-dip element of 30° to east applied. (f) DMO corrected stacked section with cross-dip element of 40° to east applied. See text for interpretation of marked reflections and diffractions. The surface location of the Barlow fault is presented on top of the section.


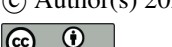



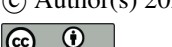

**Figure 9:** CDMO analysis for a part of the south survey around the Guercheville fault. (see Fig. 4 for the location). (a) DMO corrected stacked section with cross-dip element of 30° to west applied. (b) DMO corrected stacked section with cross-dip element of 12° to west applied. (c) DMO corrected stacked section with no cross-dip element applied. (d) DMO corrected stacked section with cross-dip element of 10° to east applied. (e) DMO corrected stacked section with cross-dip element of 30° to east applied. (f) DMO corrected stacked section with cross-dip element of 40° to east applied. The surface location of the Guercheville fault is shown on top of the section. See text for interpretation of marked reflections.





In the Chibougamau area, our strategy adjusted DMO and PSTM to find an offset range that better serves the concept of the regularity. We performed detailed velocity analysis to design a velocity model producing the highest illumination. The DMO

and PSTM images with offset range of 0-3 km provided the most convincing images for both profiles when considering only reflection coherency (Figs. 5a-b and 6a-b). Artefacts in the form of subhorizontal features appear in DMO sections where the longer offsets (0-6 km, and 0-9 km) are used to create the images (Figs. 5c, 5e, 6c, and 6e). Such artefacts disguise the DMO images of the surveys, especially in the upper crust in depths less than 6 km, and indicate a destructive contribution of CMPs in the DMO process as previously recognized in other surveys acquired in crystalline rock environments (Cheraghi et al.,

2012). PSTM images of the both profiles (Figs. 5b, 5d, and 5f and 6b, 6d, and 6f) had less capability to image steeply-dipping reflection at depths less than 6 km. This could relate to either a lack of a detailed velocity model or an inadequate contribution of CMPs especially for longer offsets. PSTM images of longer offsets do show an adequate capability of preserving deeper reflections, for example reflection chn6 in Fig. 5d and 5f (c.f., Fig. 5c and 5e, respectively) and reflections chs5 and chs6 in Fig. 6d and 6f (c.f., Fig. 6c and 6e, respectively).

## 6.2 Seismic interpretation in Chibougamau area

Both surveys imaged several packages of reflections from the near surface down to 12 km (upper crust, Figs. 5 and 6). As noted before, DMO stacked migrated sections and PSTM images with an offset range of 0-3 km presented more coherent reflections, thus our interpretation used the images shown in Figs. 5a-b and 6a-b, respectively. The geometrical attributes of the reflections are shown in Table 3. The geological map (Fig. 1) shows several fault zones in the Chibougamau area intersected

by each profile. Both profiles show a reasonable correlations of seismic reflections to the surface geology at depths less than 6 km. This helped us to map the major fault zones and interpret the seismic sections. The CDMO analysis also served as a tool to investigate out-of-plane apparent dip of the reflection packages. The interpretation of each seismic profile follows.

### 6.2.1 Seismic interpretation along the north profile

Migrated sections of the north profile (Fig. 5) show a general trend of south dipping reflectors without any conflicting dips in

the upper crust (depths less than 6 km). The contact of the Bruneau Formation (mafic volcanic rocks) with Opémisca Group (sedimentary rocks) and Obatogamau Formation (mafic to intermediate volcanic rocks) is the major cause of the reflectivity in the upper crust (chn1, chn2, chn3, and probably chn4 in Fig. 5). The reflection chn4 lies within a seismically transparent zone and also separates the deeper subhorizontal reflections sets (chn5 and chn6, Fig. 5) from the upper crust steeply dipping reflections. The thickening of the upper crust rocks around the reflection set chn3 correlates the Barlow fault and the regional

Waconichi syncline cored by a successor (Opémisca) basin (Fig. 5) (Matthieu et al., 2020b). The imaged diffraction around/within chn3 enhances its interest for mineral exploration because diffractions can associate with orebodies (Malehmir et al., 2010).





Reflection chn1 (Fig. 5, Table 3) at CDP 1300 projects to the surface within the sandstones and conglomerates of the Opémisca
Group and may correspond to internal structure such as an unconformity or small fault associated with the Waconichi Tectonic
Zone. Reflection chn2 (Fig. 5, Table 3) correlates with local structure in outcrops of Opémisca Group rocks.

At CDP 1950 reflectors within chn3 (see Table 3 for geometric attributes) correlate to the contact between sedimentary rocks
of Opémisca Group and mafic lava flows of the Bruneau Formation. This contact was mapped as the Barlow fault at surface
(Sawyer and Ben, 1993) and the migrated images (Fig. 5a-b) suggest that the fault dips at 30° to the south (Table 3). Reflectors
within chn3 also correlate with the contact of the Bruneau Formation (mafic rocks) and Obatogamau Formation (mafic to
intermediate lava flows) at CDP 2400. We previously noted that the reflection package chn3 forms the most coherent package
along the north survey in the upper crust. The CDMO analysis around reflections chn3 (Fig. 8) would suggest a 0°-10° strike
towards the east (Fig. 8c and 8d, Table 3). These reflections became weakly imaged assuming CDMO toward west (Fig. 8a
and 8b) or toward the east at dips greater than 10° (Fig. 8e and 8f). Thus reflection set chn3 most likely originates within a
complex structure off the plane of the north profile. The CDMO analysis indicates an eastward apparent dip for other upper
crustal reflection packages of the north profile (chn1 and chn2, Table 3).

Unless the north profile were extended beyond the CDP 2600 (Figs. 1 and 5) we cannot be sure that the reflection set chn4
correlates to surface geology. The regional survey in the Chibougamau area (Mathieu et al., 2020b) does not show any surface
correlation to these reflections at depth. The CDMO analysis did not show any prominent cross-dip element for this reflection
(Table 3). Deeper reflection packages (greater than 6 km) do not correlate to surface geology; subhorizontal reflections chn5
and chn6, at depths of 7-12 km, have no clear geological interpretation. These reflections show westward cross-dip element
(Table 3). Mathieu et al. (2020b) suggested that reflectors at those depths in northern Chibougamau lie within the gneisses of
the Opatica Subprovince.

The DMO stacked section of the north survey and CDMO analysis also provided insight into the diffractions within the upper
crust. Diffractions could be generated from spherical/elliptical (ore) bodies within fault zone structures and thus potentially
relevant to mineral exploration (Malehmir et al., 2010; Cheraghi et al., 2013; Bellefleur et al., 2019). Our analysis suggests the
utility of considering DMO stacked sections with cross-dips to image diffractions better.

 CDMO analysis revealed a more coherent image of the diffraction chn_diff assuming a cross-dip of 12° to west (Fig. 8b and
Table 3). In contrast, a shallower diffraction appears clearer with no cross-dip element (dashed area in Fig. 8c); this diffraction
is not imaged in the migrated section (Fig. 5a) mainly because its low amplitude did not survive a migration that collapsed
diffraction energy. In order to scrutinize the diffraction imaging capability, we compare an enlarged section of the upper crust
of the Chibougamau north survey (shallower than 5 km) with no cross-dip applied (Fig. 8c) with a section with cross-dip 12°
to the west applied (Fig. 8b) in Figs. 10 and 11, respectively. Figure 10a clearly shows the diffraction tail imaged within





reflection package chn3 at CDP 1600 (marked with red dashed ellipse). Normally, the signal energy generated from diffractions appears weaker than those of reflections and the processing flow further enhances the S/N ratio in favor of the reflections;

diffractions are easy to miss so that a focused visual inspection is necessary (Malehmir et al., 2010; Cheraghi et al., 2013;). The marked diffraction in Fig. 10a shows approximately 1 km lateral coherency between CDPs 1500-1700; to evaluate signal associated within this diffraction, we carefully inspected the processed shot gathers (Table 2 for the processing steps applied) along the survey at CDP locations where the diffraction appears. Figure 10b shows shot gather 4070 (see Fig. 4a for location) around CDP 1600. The reflection chn3 appears in this shot gather and also diffracted waves at times less than 1 s.


A zoomed view of the diffraction chn_diff in a section with a cross-dip element of 12° to west is shown in Fig. 11. Similar to the analysis shown in Fig. 10, we visually checked the shot gathers around CDP locations where chn_diff was imaged (CDPs 1900-2200). Shot gather 2730 (Fig. 4a for location) is shown as an example. This shot gather imaged a package of reflections interpreted as chn3 and also diffracted events at approximately 1.5 s in CDP locations where chn_diff was expected to be

imaged (see CDP 2088 marked as the apex of the diffraction in Fig. 11b).

### 6.2.2 Seismic interpretation along the south profile

The south profile shows more complexity in the upper crust where both north and south dipping reflections are imaged (Fig. 6). It seems that the lithological contact of the Obatogamau Formation (intermediate to mafic rocks) and the Caopatina Formation (sedimentary rocks) is the main cause of the reflectivity along the south profile in the upper crust (Fig. 6). The

volcanic-sedimentary reflection packages in the upper crust (chs1, chs2, and chs3) and deeper reflection packages (chs4, chs5, chs6) depict a synform structure along the south profile. The geometry of this structure includes the south dipping reflection in the north of the profile and north dipping reflection in the south (Fig. 6). Similar to the north profile (Fig. 5), the upper crustal rocks around the reflection sets chs1, chs2, chs3, and chs4 (Fig. 6) are approximately 6 km thick. The correlation of these reflections with the Guercheville fault and the Doda fault zones (Figs. 1 and 6) could suggest potential metal endowment.


Reflection chs1 (Fig. 6, Table 3) at CDP 1700 likely correlates with the contact between pelitic to siliciclastic sedimentary rocks of the basin-restricted Caopatina Formation and mafic to intermediate lava flows of the Obatogamau Formation. This reflection set does not show any cross-dip element towards east or west (Table 3).





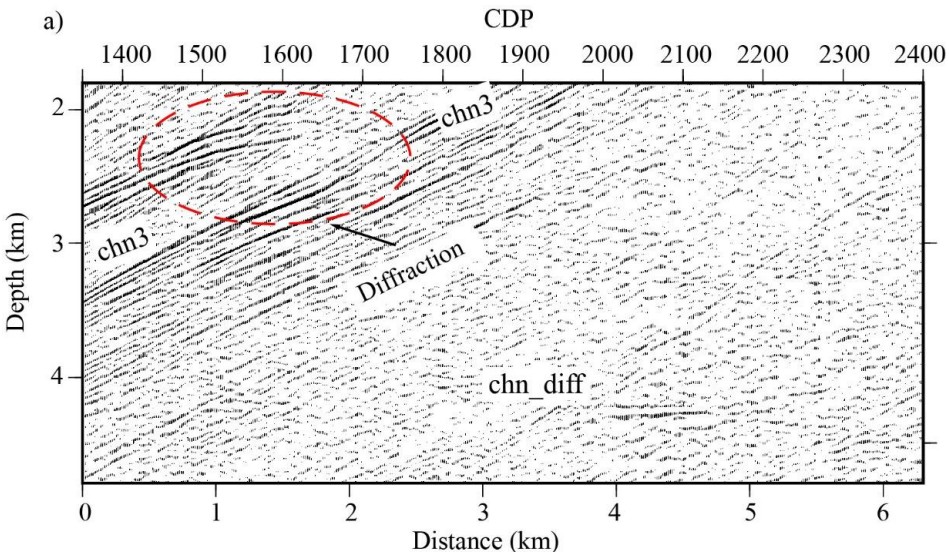

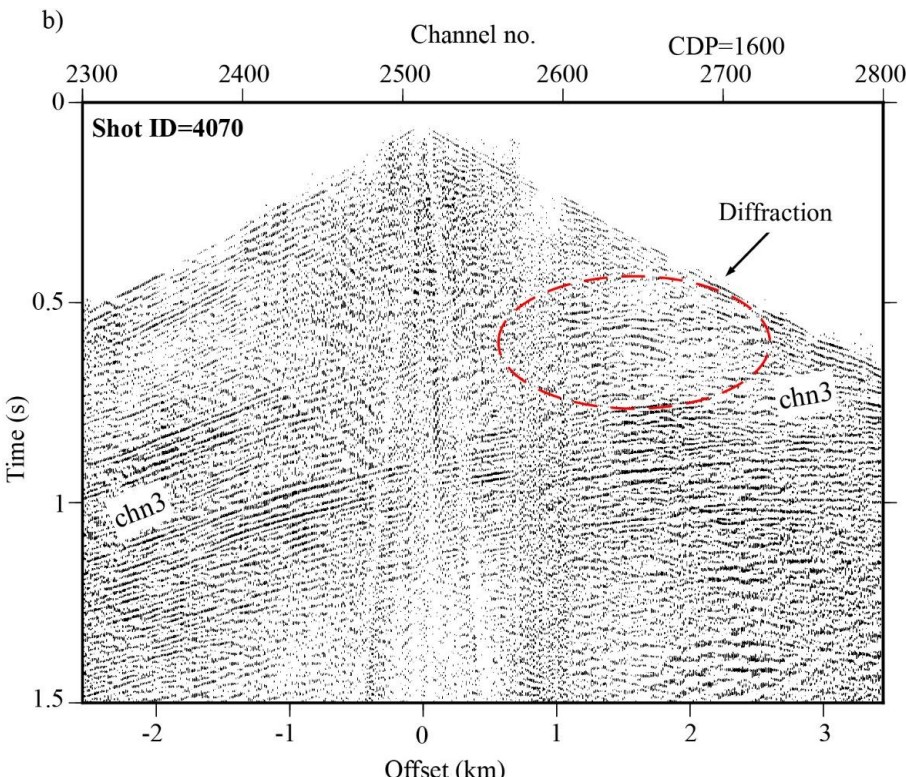

**Figure 10**: (a) A zoomed view from Fig. 8c (DMO stacked section with no cross-dip element applied) around the diffraction imaged. (b) Shot 4070 (see Fig. 4 for the location) acquired for the north survey which shows the signal from the diffraction around CDP 1600 in (a). The location of CDP 1600 is shown in (b). See text for interpretation.







**Figure 11:** (a) A zoomed view from Fig. 8b (DMO stacked section with cross-dip element 12° to west applied) around the diffraction chn_diff. (b) Shot 2730 (see Fig. 4 for the location) acquired for the north survey which shows the signal from the diffraction chn_diff; the apex of chn_diff is imaged around CDP 2088 in (a). The location of CDP 2088 is shown in (b). See text for interpretation.






Reflection sequence chs2 (Fig. 6, Table 3) also correlates with the contact between the Obatogamau (sedimentary rock) and
Caopatina Formations (mafic rocks), but includes two packages of reflectivity including a set of steeply dipping reflections
and another set of subhorizontal reflections (Fig. 6). The surface geology above the subhorizontal set of chs2 contains mafic
rocks of the Obatogamau Formation. The surface location of the Guercheville fault is marked at CDP 2400, thus the reflection
set of chs2 could be associated. The Guercheville fault is locally measured as subvertical (Daigneault, 1996). The reflection
chs2 has a 40° dip to south in the migrated section (Fig. 6 and Table 3), which is in much less than the reported field
measurements. Further knowledge about the geometry of reflection chs2, if associated with the Guercheville fault, would help
to better understand the subsurface architecture and its relationship to gold deposits along strike to the east. CDMO analysis
along the south survey (Fig. 9) suggested dips for reflection chs2 varying between 20°-50° depending on different CDMO
correction values. To evaluate CDMO results around chs2 shot gather 14135 is considered. Figure 12 shows shot gather 15135
from the south survey (see Fig. 4d for location) that was acquired near CDP 2220 where chs2 turns from a steeply-dipping
reflector into a subhorizontal reflector (see Figs. 6 and 9). The chs2 reflection in this shot gather shows both subhorizontal and
steeply-dipping parts at approximately 1 s (see the dashed line in Fig. 12, which separates those parts). The steeply dipping
part of chs2 in Fig. 12 has an associated high apparent velocity (~ 8000 m/s), required so that a reflector dipping ~ 40°-50°
constructively stacks; this appears consistent with Fig. 9c (no cross-dip applied) and sections with cross-dip element to east
(Fig. 9d, 9e, and 9f). These reflections are also imaged with westward CDMO (Fig. 9a and 9b). This uncertainty would suggest
greater complexity of the Guercheville fault off the plane of the south profile.

Similar to reflection sets chs1 and chs2, the reflection set chs3 (Fig. 6, Table 3) correlates with the contact of the Obatogamau
and Caopatina Formation at CDP 500. Unlike the reflection sets chs1 and chs2, the chs3 set dips to the north (30°, Table 3)
and represents the deepest reflector associated with the contact of the Obatogamau and Caopatina formations along the south
survey (Table 3). The CDMO analysis implies that the north dipping reflector chs3 shows more coherency with westward
strike (12°-30°, Fig. 9b and 9a, respectively). The reflector chs3 is less coherent at depths shallower than 2 km. This may
suggest a steeper dip that CDMO was not able to image.

Reflection chs4 (Fig. 6, Table 3), located at depths of 2-5 km, dipping towards north with a westward cross-dip element,
probably lies within mafic rocks of the Obatogamau or Waconichi formations; therefore, it most likely originates at more felsic
interlayers, chert and iron formations, sulphide (VMS) accumulations, or faults within the mafic rocks. If interpreted as a fault,
reflection chs4 most likely correlates to the Doda fault. The Doda fault is measured subvertical at surface (Daigneault, 1996).
The absence of reflectivity at depths less than 2 km on top of the reflection set chs4 could result from these steep dips and
limited survey offsets.


At depths of 6-9 km, two packages of subhorizontal reflections, chs5 to north and chs6 to south (Fig. 6, Table 3), are appropriate
for the proposed basal contact of the greenstones with underlying tonalite-trondhjemite-granodiorite (TTG) or tonalite-



trondhjemite-diorite (TTD) intrusive rocks (Mathieu et al., 2020a). If these reflection sets are generated inside the intrusive

rocks, then they could show the reflections from felsic rocks of the Hébert pluton. (Mathieu et al., 2020b). The extension of

these felsic rocks along the south profile, underlain with reflection set chs3 and chs4 (north dipping faults, Fig. 6, and Table

3) in the south and chs2 (south dipping fault, Fig. 6, Table 3) shows thickening of the upper crustal rocks, perhaps associated

with the regional Druillettes snycline (Mathieu et al., 2020a).

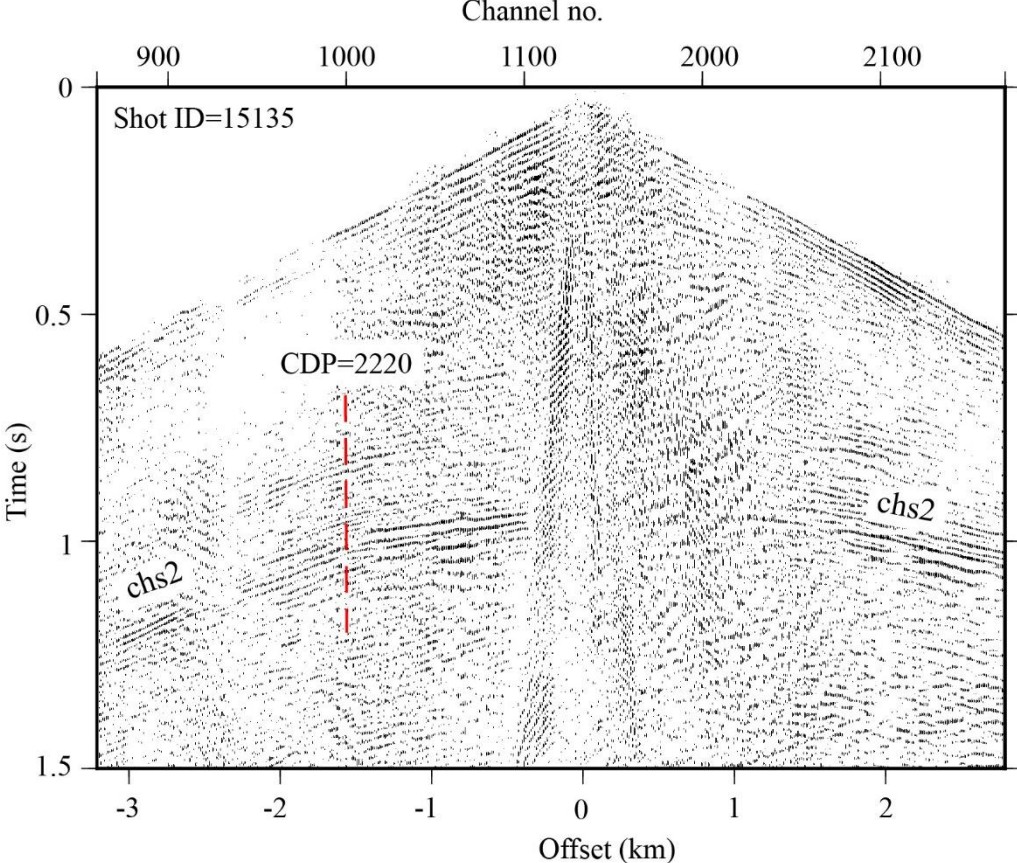

**Figure 12:** Shot gather 13135 acquired for the south survey (see Fig. 4 for the location). A package of reflections interpreted as chs2 in Fig.

6 is imaged in this shot; The location of CDP 2220 is marked (see Figs. 6 and 9 for the location) and is marked on the shot. This CDP location

shows separation of subhorizontal and steeply-dipping part of chs2. See text for interpretation.






### 6.3 Potential for exploration of orogenic gold

The Barlow fault and the associated diffractions in the north and the joint complex structure of the Guercheville fault in the south and the Doda fault all are imaged within the greenstone belt rocks of the upper crust (Mathieu et al., 2020a). Both surveys show deep reflectors, the reflections chn5 and chn6 along the north profile and the reflections chs5 and chs6 along the south

profile, that appear related to regional synclines. The faults favor locations on the margins of synclines and appear related to late deformation (folding) of the successor basins that core the synclines. Orogenic gold (Au) systems typically require major (crustal-scale) faults to channelize fluids, and steep or sub-vertical faults are more efficient at doing that. The three faults imaged and discussed here continue to 15-20 km depths (Mathieu et al., 2020b). They may not reach deep enough to channel deep-sourced Au-bearing fluids/magmas, but may localize and enable small volumes of magma to rise toward the surface from

the mid- to lower-crust. Numerous Au-showings along the Guercheville fault east of the seismic profile indicate that some faults do localize Au-bearing fluids or magmas.

### 7 Conclusions

Analysis of high-resolution seismic profiles in the Chibougamau area revealed the crucial role of survey geometry on seismic illumination. Seismic data processing steps such as DMO corrections and PSTM proved to be highly dependent on a regular

offset distribution of CMPs in CDP bins for their effectiveness and also an optimized offset range that provides better illumination in the presence of a complex subsurface architecture. The regular distribution of CMPs directly affects the performance of DMO and PSTM algorithm. A detailed velocity model could also increase the illumination when a DMO or PSTM algorithm is utilized. The key step for optimized DMO and PSTM processing is the investigation of offset distribution in order to choose an offset range in which most of the CDP bins show regular distribution and thus contribute better to each

process. We specifically investigated this for two high-resolution seismic surveys with offsets in a range of 0-10 km and the analysis indicated that an offset range of 0-3 km provides more regular sampling. Further investigation performed on the common-offset DMO correction process and common-offset PSTM for the entire available offset range of 0-10 km (at a step rate of 3 km) indicated that both profiles showed their best results for the offset range of 0-3 km. This offset range also provides the better illumination for DMO and PSTM.


The subsurface architecture in the Chibougamau area has complex structure within its fault systems, these fault systems potentially correspond to gold endowment and thus provide a major motivation for the survey and the processing trials. The comprehensive processing workflow applied in this study improved the imaging of several major faults in the area. The crooked nature of the surveys encouraged performing CDMO analysis to take into account the effect of out-of-plane structures. The

seismic imaging revealed the general trend of south dipping structures including the Barlow fault along the north survey to depths of 15 km. The CDMO-DMO stacked sections imaged some diffractions along the north profile within the reflection package associated with the Barlow fault. The seismic image also shows the thickening of the upper crust rock beneath the




Barlow fault within the regional Wachonachi syncline. The seismic imaging along the south profile shows a more modest thickening of the upper crustal greenstone and metasedimentary rocks around reflections associated with the Guercheville and

Doda faults. The seismic image shows a regional synform structure along the south profiles. The Guercheville fault relates to south dipping reflectors on the north limb of the Druillettes syncline and numerous gold showings along its strike. The DMO-CDMO results indicate a complex structural fault geometry. The Doda fault projects to a north dipping reflector, but this fault is not imaged at depths of less than 2 km.

## 8 Appendix A: evaluating survey geometry for DMO and PSTM

For a 3D survey, equal azimuthal distribution, typically contributed by inline and crossline components, satisfies the symmetric sampling (Vermeer 1990, 1998, 2010). In the case of a 2D survey, reciprocity of shot/receiver gathers suggest that properties of the continuous wavefield in a common shot/VP gather are the same as the properties of a common receiver gather. Sampling requirements are the same for both domains and results in symmetric sampling. The immediate requirement of the 2D symmetric sampling is that the continuous wave field should be alias-free for ground-roll and low velocity noise (Vermeer,

2010). To satisfy an alias-free, continuous wavefield sampling, the basic sampling interval ($\Delta x$) is defined as Eq. (A1) (Vermeer, 2010):

$$\Delta x = \frac{V_{min}}{2f_{max}} \tag{A1}$$

where $V_{min}$ is the minimum apparent velocity and $f_{max}$ is the maximum frequency of data. The VP and receiver spacing for high-resolution surveys in the Chibougamau area are 6.25 m and 12.5 m, respectively (Table 1). For a representative shot

gather (receiver spacing of 12.5 m) and an estimated maximum frequency range of 60-120 Hz, the minimum apparent velocity would be 1500-3000 ms$^{-1}$, and for a receiver gather with shot spacing of 6.25 m the minimum apparent velocity would be 750-1500 ms$^{-1}$. These calculated apparent velocities indicate that the Chibougamau profiles are alias-free regarding shear waves and ground roll.

The basic signal sampling interval ($d$) required to acquire a desired part of the continuous wavefield, (i.e., P- wave energy) alias-free can be defined with Eq. (A1) and $V_{min}$ is the minimum apparent velocity in the signal part, e.g., 5000-5500 ms$^{-1}$ for a typical crystalline rock environment. Assuming these velocities, the receiver and VP spacing in Chibougamau profiles are much smaller than the basic requirement and the acquired signal is alias-free for P-wave energy. The benefit of acquiring alias-free signal for receiver /VP gathers is that those gathers act as an anti-alias filter for remaining low velocity noise (e.g., 300-

1500 ms$^{-1}$ in Chibougamau profiles).

Acquiring a seismic survey on the planned shot and receiver locations is not always practical due to natural obstacles or economic considerations. Gaps result in missed shots/receivers and sparse CMP distribution for some locations, or acquiring extra shots in other places with a resulting coarse CMP coverage. The crooked geometry exacerbates the effect of improper

 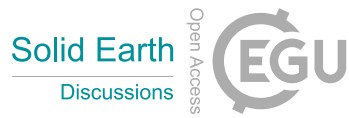

CMP distribution. The irregularity of a survey is defined as sparse CMP distribution in some parts of the survey and overabundance of CMPs in other parts (Beasley and Klotz, 1992). Some of the essential multichannel processing steps, and especially wave equation processes such as Kirchhoff PSTM and/or DMO corrections, assume that shots and receivers were acquired in nominal places and that a continuous CMP coverage (regular geometry) was fulfilled. The irregular geometry may lead to artefacts or footprints for PSTM and DMO process (Canning and Gardner, 1998; Schuster and Liu, 2001). The effects

of those artefacts on Kirchhoff PSTM algorithms and DMO corrections can be defined basically as a concept of an integral summation (Canning and Gardner, 1998):

$$f(x,y,z) = \int w \frac{d}{dt} f(S,R,\tau) dSdR \tag{A2}$$

$S$ and $R$ represents shot and receiver coordinates, respectively; $(x, y, z)$ is a diffraction point $(p)$ and $\tau$ is traveltime along the diffraction surface generated by $(p)$. When common-offset gathers are considered for PSTM algorithms or DMO corrections,

$dSdR$ will be the CMP coordinate, i.e. $dx_m dy_m$ where $x_m$ and $y_m$ are CMP coordinates and offset planes are shown by $w$. For a regular geometry offset increments are constant and thus we can assume that $dx_m dy_m$ is constant and offset planes $(w)$ including short and long offsets contribute equally in the Eq. (A2). In a case of irregular geometry, CMP locations (i.e. $dx_m dy_m$) and $w$ (i.e. offset planes) will contribute irregularly in the Eq. (A2). For a Kirchhoff style PSTM if CMPs are irregularly distributed (per their offsets), the migrated traces would destructively contribute in the stacking process and the

resulting seismic image will be blurred (Yilmaz, 2001). For DMO corrections, an imaging point represents a contribution of CMPs for both short and long offsets in the DMO formula (Deregowski, 1982). If some of the offsets are missing around the imaging point, the DMO process generates artefacts (Vermeer, 2012), generally in the form of subhorizontal features that disguise the seismic image (Cheraghi et al., 2012).

To further investigate the effect of regular offset plane for DMO corrections, we generated an example of common-offset DMO corrections which is shown in Fig. A1 based on the seismic wave velocities typically observed in crystalline rock environments. The graph has been provided from DMO formula (Hale, 1991) with considering common-offset method (Fowler, 1998). This graph implies that the missing offsets (i.e., irregularity) hinder the DMO correction process, i.e., the curve will be discrete.


The above mentioned irregularity of the wave equation processes and its effect has been subject of many studies (e.g., Williams and Marcoux, 1989; Ronen, et al., 1995;). The less studied subject is CMP contribution into subsurface illumination of those processes (e.g., DMO fold, Vermeer, 1994; Ferber, 1997). The conventional CMP stacking fold is defined based on total number of traces sharing a reflector point on a flat surface. All these traces contribute to the subsurface illumination (Beasley

and Klotz, 1992; Beasley, 1993; Ferber, 1997). The standard CMP stacking can also be applied to single-dip reflectors, if dip-dependent velocity i.e., apparent velocity, is considered (Jakubowicz, 1990). Cases of lateral velocity changes, diffractions, and conflicting dips require more advanced processes. The pre-stack depth migration is the solution for the first and the others

need DMO or PSTM to be applied (Jakubowicz, 1990). For a particular reflector with an arbitrary dip and strike the DMO fold (or DMO illumination) is considered to be those traces that contribute to the process constructively (Ferber, 1997). For a given

source and receiver location, constructive DMO illumination takes place if the difference between DMO and NMO corrected travel-time reflection and zero-offset travel-time reflector is less than half of the dominant wavelength (Ferber, 1997). In the best case scenario, DMO fold is equal to CMP stacking fold (Vermeer, 2010). The DMO illumination can be investigated during survey design with numerical modeling of seismic response where different scenarios are considered for subsurface architecture (Beasley, 1993). For the acquired geometry, the regularity of CMPs is the most crucial factor which defines the

optimized performance of any wave equation process (DMO and PSTM, Canning and Gardner, 1998).

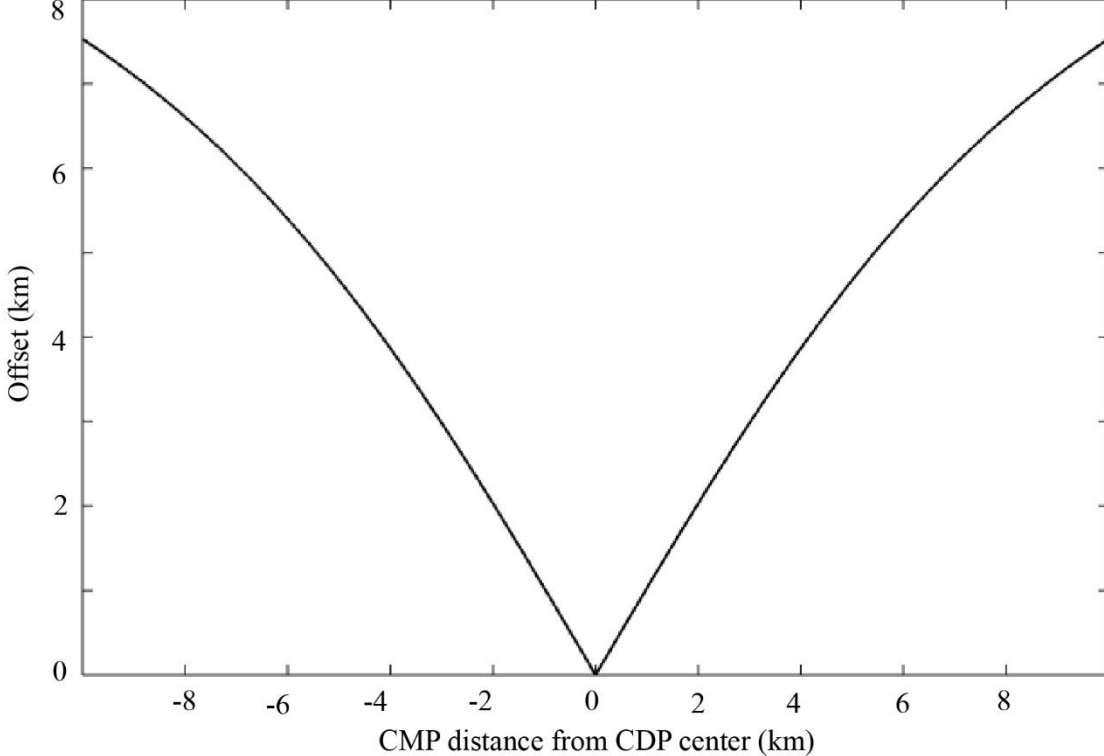

**Figure A1:** The regular offset distribution in a CDP bin for DMO corrections calculated from DMO formula (see Hale, 1991; Fowler, 1998). The offset range is considered 0-8 km; the average velocity is considered 5500 ms$^{-1}$ to be representative of crystalline rocks. The recording length is 4 s with sampling rate of 2 ms (similar to Chibougamau high-resolution seismic surveys, see Table1). Target depth is located at 1s.






## 9 Acknowledgments

This research was funded by the NSERC Canada First Research Excellence Fund. The authors would like to thank the Metal Earth project at Laurentian University for providing and archiving the seismic data. S. Cheraghi acknowledges Metal Earth for funding his research. Globe Claritas was used for seismic processing. GMT from P. Wessel and W.H.F. Smith was used to
prepare some of the figures. GOCAD was used for 3D visualization and interpretation. The authors would like to thank Kipp Grose for IT support during processing of Metal Earth seismic surveys. Dean Meek is acknowledged to have provided geological and geographical maps in the study area. The authors would like to appreciate frontline and essential workers who risk their lives during pandemic spread of the COVID-19. This is a collaboration of Metal Earth and Smart Exploration. Smart Exploration has received funding from the European Union's Horizon 2020 research and innovation program under grant. This
is Metal Earth publication MERC-ME-2020-093.

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
