# Peer review of "An analysis of pre- and post-stack migration approaches in the"

_Solid Earth, 2020_

## Referee Comment (RC1) · Fomin Tanya (Referee) · 16 Nov 2020

General comments:

1. Well written paper with a clear objective and straightforward structure. This case study example is very useful and practical for seismic processors. In these days PSTM and PSDM methods become most popular and DMO technique called "old fashioned" not used broadly anymore by commercial processing companies (even some seismic software not include DMO in their packages). 2. I think one of the difficulties of this

topic that you cannot provide "a recipe" what would be the best technique DMO, PSTM or something else for particular geological environment until you test it and apply all possible methods. That is not very practical. It would be good at least if you provide some recommendations on possible processing flows for different geological environments for example DMO should work for some areas and not really useful in others. 3. It is a very detailed interpretation section of seismic reflectivity which is very good but could be over interpreted.

Line 20 – methods instead of method Lines 21-22 – What was a reason of 3km increment and was a step 3km or you checked as well 2-4km, 3-5km etc? Would be 0-3km offset recommendation or it has to be checked for every seismic survey. Line 27 – From the Figure 1 it looks like Profile just stops before the Doda fault and not crossing the fault.

Line 54 – Cheraghi et al., 2012 is not on the list of References Line 64 – Bellefleur et al., 2018 is not on the list of References

Line 84 – David et al., 2011 is not on the list of References Line 103 – Dinmroth et al., 1995 spelling and is not on the list of References Line 108 – Daigneault and Allard, 1990 is not on the list of References Line 111 – Bedeaux et al., 2020 is not on the list of References

Line 127 – How is significant to have more denser VP instead of receiver spacing (cost is more for shots not for channels)

What is a Moho depth? Why is only 12 sec record length? Was any testing for higher ending frequencies 150Hz or even higher? Line 143 – Common lower case Line 169-170 – "We designed offset . . . " Was this designed only based on visual assessment or something else?

Line 221 – You don't need to have "The distribution. . ." sentence second time. In the Figure 4 offsets 0-3km, 3-6km and 6-9km. In the text, it is 0-3km, 0-6km and 0-9km.

Am I wrong of reading that?

In Table 2 First arrivals picked up to 10km. Is any need for that? I assume this is a one-layer refractor model? Why top muting but not just stretch with some %?

Line 251 – Why was used a constant velocity for DMO corrections?

Figures 5 and 6 should be Depth converted migrated sections?

Line 385 – See capital

Figure 9 You need better arrow for the fault location (similar to figure 8)

Line 518 - shot gather 15135 but in the figure 13135

Line 648 – Vermeer, 1994 is not on the list

Line 652 – 653 " The pre-stack depth migration . . ." something missing in this sentence?

Line 697 – Is it 2018? Line 738 – I could not find this reference in the text.

Final remarks

It is a good and useful paper for people who process seismic data particular for hard rock data sets. We need to be very careful and not to over interpret reflection seismic data by trying to fit to geological model.

---

## Author Comment (AC1) · 19 Nov 2020

Dear Reviewer,

First We would like to commend your inclusive review and the detailed comments you provided. Your comments definitely increase the quality of the paper. We briefly reply your comments here and in the revised manuscript we address them in details.

About your general comments on application of DMO, PSTM and PSDM: We agree that more advanced computing systems have facilitated the application of the sophisticated

methods such as PSTM and PSDM in seismic processing, in general. As you also mentioned it is not possible to say which method is the best for hard rock environment before it is tested. Our goal was to compare the conventional processing method (DMO stacked migration) with more advanced method (PSTM) where the survey is crooked. In the revised version we provided some recommendations on possible processing flow, as you mentioned, to address the challenges in hard rock seismic processing. The offset step rate of 0-3 km, 3-6 km, and 6-9 km is designed based on the distribution of CMPs for the acquired geometry in Chibougamau area. The offset step rate has to be chosen based on the geometry and could vary for each specific survey.

In the revised manuscript we also consider your specific comments on references, typo, ...

Best regards, Saeid Cheraghi

---

## Referee Comment (RC2) · Anonymous Referee #2 · 17 Feb 2021

**Comments on Seismic imaging across fault systems in the Abitibi greenstone belt – An analysis of pre- and post-stack migration approaches in the Chibougamau area, Quebec, Canada, by Saeid Cheraghi et al**

This paper discusses the processing strategy to apply to vertical incidence seismic reflection data in order to get best subsurface image along crooked-line acquisitions. In that regard, I think the paper is excellent. The analysis it makes of the importance of CDP bin centers in relation to CDPs location and maximum offset, together with the need of undertaking cross-dip move out tests is crucial to get the best resolved image. However, I'm a bit disappointed about the interpretation of the data. I don't know if the goal of this paper is to present also a geological model of the area, as they don't really do it. At this point, I don't see a clear relationship between the faults and the reflectivity. As an example, the Doda fault does not seem to have a seismic response in the southern profile. The width of the reflectivity associated to the Guercheville and Barlow faults is much higher than the trace of the faults themselves. So what is the reflectivity responding to? Deformation or lithological contrast? Is the later related to Au mineralization or not? In relation to this, the map in figure 1 lacks information about dips and a cross-section where we can have an idea of the structure. Finally, to round up the conclusions, both profiles should be plotted overlapping the entire seismic profile, presenting the overall structure of the area. As it is now, the paper is a good technical work with limitations regarding the interpretation. Finally, even though there are native English speaking researches among the authors, I find the text awkward sometimes. Conclusions read like a telegram, and some parts of the text do not flow properly. So a revision of the grammar and style would be convenient (from my point of view).

Ahead I give more details about the points made above. I also add some other minor comments

Line 16-17: Would help to know what type of faults are them. Strike slip, normal or reverse? As there is not a proper stratigraphic column in the legend, the kinematics of the faults is difficult to infer

Line 23: In the northern?

Line 24: structures or the key geological structure….

Line 83: Fig. 1 does not show …the NE portion of the Abitibi super-province. A regional scale map (at least as the inset in figure 1) where this province appeared will be much better.

Line 84: Same applies for the ages of rocks. Include them in the legend of Figure 1 and not only in the text  so we can have a quick idea of the structure. Right now we don't know which ones are older or younger.

Line 252: …various constant velocities between 5000-6500 m/s, with a step range…..?

Line 258. Here you start presenting results but the whole thing is quite messy. A new paragraph (add inter-paragraph space as you do in other places, e.g., between lines 295 and 296) should start in line 258 and another one in line 264 (The design of the north….)

Line 268: ….Labelled in Fig. 5, chn1……….

Line 271. Those reflections project to the surface off the CDP line, so rephrase. Is not that they show no correlation with the surface geology. We don't see it. And the map you provide in Fig 1 lacks detail, but some could be related with the Burlow pluton southern boundary?

Line 282: I don't see chs5 and6 as subhorizontal. If something, chs5 has a hyperbolic geometry with high opposite dip to ch6 in its deeper part

Line 272:….one kilometer…you are using km and numbers so it should be 1 km.

Line 276: New paragraph again (interparagraph space)

Line 284: I'd like that title to be more specific.

Line 289: Change to……When out-of-place CMP's scattered/reflected seismic waves from steep structures off the CDP line (cross-dip direction) exist, cross-dip analysis addresses…..??As it is now, the phrase does not read well. I think you make an excesive use of semi-colon when you could replace it by commas or new paragraphs.

Line 311: Remove…Table 3 shows…..segment. It is already mentioned in the previous paragraph

Line 312: Remove…Table 3 shows…..segment. It is already mentioned in the previous paragraph.

Line312:  Remove…. The DMO-CDMO stacked sections are essential for the diffraction imaging. This is not the place for that assessment.

Line 314:……depths lesser than…..

Line 322: …40º to the south and features lesser continuity (Fig. 9c). You should also take into account the continuity of the reflection

Line 356: High resolution seismic profiles….

Lines 435: Unconformities are identified in vertical incidence data when the reflections they truncate are visible???. In my opinion it is more likely that chn1 responds to lithological variations inside the Opemisca Group.

Lines 436: This interpretation of chn2 is incomplete. What is the structure of the Opemisca Group there? Why doesn't it respond to another change in lithology?

Line 443 and fw: Rewrite, as there are articles missing. For instance:
The CDMO analysis around reflections chn3 (Fig. 8) would suggest a 0°-10° strike towards the east (Fig. 8c and 8d, Table 3). Furthermore, these reflections became weakly imaged assuming a CDMO toward the west (Fig. 8a 445 and 8b) or toward the east at dips greater than 10° (Fig. 8e and 8f). Finally, the CDMO analysis also indicates an eastward apparent dip for other upper crustal reflection packages of the north profile (chn1 and chn2, Table 3).

Line 456:………provided insights….

Linbe 457:…..they are potentially relevant….

Line 456-480: You need to rewrite that part as you merely do a description, but not a discussion. So discuss, e.g., why CDMO should help in imaging diffractions given the geometry of the waves in the latter. Also, you can discuss what you think they represent. As it is now, there is no discussion in there.

Line 524-525: This is a really interesting problem. But there should be ways to address it. The south profile is oblique to the fault at the cross-point so in Figure 9c you are imaging an apparent dip. But apart from knowing the angle between the profile and the fault at that point, you also know that real dips are higher than apparent dips. So probably the best image of the fault is that where its reflection looks shorter and steeper. This should help you to provide the real geometry associated to that feature. Moreover, addressing your preferred image of the fault in figure 9 is part of what a discussion should be, instead of just saying that the fault has a complex geometry (something that is not clear from the map, as it looks subvertical and simple).

Lines 531: What is the dip of rocks at the surface??? This implies again the need of a geological map with layer dips and a cross-section.

Line 536: If interpreted as a fault, reflection chs4 most likely correlates to the Doda fault?. But Doda fault projects at CDP 100 and these reflections project outside the profile! Even if dip changes and the fault becomes subvertical, you are imaging things at 2 km in the migrated section and nothing at the surface where the Doda Fault projects. It seems unlikely that chs4 represents that fault. Finally, at the cross-point with the Doda Fault, the profile is oblique, so you are seeing apparent dips. Does Figure 9 provide better insights about this fault? This should be better discussed.

Line 541: Those reflections are not subhorizontal. They have opposite dips and suggest a syncline structure.

Lines 557-558:…. joint complex structure of the Guercheville fault as well as the Doda fault in the south are all imaged within the greenstone belt rocks of the upper….

Line 559:…deep reflections: chn5 and chn6…..Do not mix reflectors and reflections.

Conclusions: I don't see the Doda Fault anywhere in the seismic section. It projects around CDP 100 in the seismic profile and there, the strongest reflectivity is at 2-3 km. You need to discuss that in the discussion part, but it is not a clear conclusion with the present discussion.

Table 2: Is it necessary to be that specific in step 6? Is not enough to say $V^2t$?

Does the time to depth conversion use velocities higher than migration velocities? Explain or change

Figure 1: Geological map should have dips. Furthermore, a cross section along the profile should be presented. I'm sure there are has been plenty of structural geologists working on that.

Figure 4: Caption-…………..shot and CDP locations are also……

Figure 5, 6 etc: Add N and S in the edges of the profile. Although the reader can figure out the dip of reflections, it is faster to indicate the orientation in the profile itself.

---

## Referee Comment (RC3) · Anonymous Referee #3 · 3 Mar 2021

The manuscript presents a processing strategy to image strong dipping reflections from a crooked-line acquisition. The manuscript is well written and the methodology is well described. Although, the geological interpretation is not really developed. The manuscript should be a significant and valuable work, and it fits the scope of Solid Earth's special issue "State of the art in mineral exploration". However, some suggestions and technical comments are as follow:

1. It is not clear why the different offset ranges of 0-3, 3-6, and 6-9 km are selected, even considering that the seismic profile is 10 km. In addition, during the text, these

ranges are changing from 0-3, 3-6, and 6-9 km to 0-3, 0-6, and 0-9 km. Which is the correct one?

2. In the interpretation, some of the reflections are associated with different geological structures. However, the geological map does not provide any strike and dip information. Also, the profile, to the south, does not cross the Doda fault while in the interpretation one of the reflections is associated with the fault. In the conclusions, it is mentioned that this fault is only imaged in the first 2 km, while the chs4 (associated with the fault) is observed around 2–3 km depth. Are you sure that chs4 is the fault? In the text, it is not clear the origin of the diffractions. Is it the fault or a potential ore body?

3. It is possible to provide a geological model of the final interpretation? Which are the relationship between chn1 and chs1?

4. Can you provide the orientation of the seismic profiles in Figures 5–12? In Figure 7, why is represented 12 km, if only there are interpreted the first 6 km, why chn4-chn6 are not interpreted in this figure? Also, in Figure 10, the chn_diff seems not to agree on the shot gather and the stacked section.

5. In the text is mentioned that for DMO it is used a range velocity of 5000–6500 m/s, while for the CDMO is used a constant velocity of 5500 m/s. Why this decision was made, can you explain it further? In addition, in my understanding, a range of velocities, such as 5000–6500 m/s, will not be considered a constant velocity.

6. Some references are missing

In the attached file you can find all further comments related to the manuscript.

Please also note the supplement to this comment:
https://se.copernicus.org/preprints/se-2020-155/se-2020-155-RC3-supplement.pdf

**Supplement:**

The authors of the manuscript titled "Seismic imaging across fault systems in the Abitibi greenstone belt – An analysis of pre- and post-stack migration approaches in the Chibougamau area, Quebec, Canada" presents a processing strategy to image strong dipping reflections from a crooked-line acquisition. The manuscript is well written and the methodology is well described. Although, the geological interpretation is not really developed. The manuscript should be a significant and valuable work, and it fits the scope of Solid Earth's special issue "State of the art in mineral exploration". However, some suggestions and technical comments are as follow:

1) It is not clear why the different offset ranges of 0-3, 3-6, and 6-9 km are selected, even considering that, the seismic profile is 10 km. In addition, during the text, these ranges are changing from 0-3, 3-6, and 6-9 km to 0-3, 0-6 and 0-9 km. Which is the correct one?

2) In the interpretation, some of the reflections are associated with different geological structures. However, the geological map does not provide any strike and dip information. Also, the profile, to the south, does not cross the Doda fault while in the interpretation one of the reflections is associated with the fault. In the conclusions, it is mentioned that this fault is only imaged in the first 2 km, while the chs4 (associated with the fault) is observed around 2–3 km depth. Are you sure that chs4 is the fault? In the text, it is not clear the origin of the diffractions. Is it the fault or a potential ore body?

3) It is possible to provide a geological model of the final interpretation? Which are the relationship between chn1 and chs1?

4) Can you provide the orientation of the seismic profiles in Figures 5–12? In Figure 7, why is represented 12 km, if only there are interpreted the first 6 km, why chn4-chn6 are not to interpreted in this figure? Also, in Figure 10, the chn_diff seems not agree on the shot gather and the stacked section.

5) In the text is mentioned that for DMO it is used a range velocity of 5000–6500 m/s, while for the CDMO is used a constant velocity of 5500 m/s. Why this decision was made, can you explain it further? In addition, in my understanding, a range of velocities, such as 5000–6500 m/s, will not be considered a constant velocity.

6) Some references are missing.

Minor comments:

L.15 Canada,

L.17-18 with a known metal endowment in the area

L.24 key geological structures // sets

L.35-36 (Juhlin, 1995a; Juhlin et al., 1995 and 2010; Bellefleur et al., 1998 and 2015; Perron and Calvert, 1998; Ahmadi et al., 2013)

L. 36 add comma after "however"

L. 67 Mercier-Langevin et al., 2014

L.33  crystalline rock

L. 35-36 Juhlin, 1995a; Juhlin et al., 1995, 2010; Bellefleur et al., 1998, 2015 ... // remove ; // . However, all ...

L.37 than a coherent

L. 38 Petrophysical measurements, where available, complemented with reflectivity/velocity models of the shallow crust, ...

L. 39 permit a more

L.42 limited the application

L.45-46 confusing sentence. Can you rephrase it?

L.47 change "which" for "that"

L.59 Bedides

L.61-63 confusing sentence. Can you rephrase it?

L.78 Are the offset ranges between 0-10 km or 0-9 km? In the abstract is mentioned 0-9 km, while here is between 0 and 10 km. It is somewhat confusing.

L.90felsic lava flows

L. 92 Mueller et al., 1989; Leclerc et al., 2017

L.92-93 ..., observed along the southern profile, ...

L.102 , and

L. 103 Dimroth et al., 1995

L.106 change "to" for "with"

L. 108-109 Leclerc et al., 2012 and 2017 // schistosity

L.114 at the surface

L. 117 Dimroth, 1985; Mueller et al., 1989

L.120-121 confusing sentence. Can you rephrase it?

L. 148 low-velocity

L.142 DMO or PSTM,

L.152 artifacts

L.158 . for example,

L.161 , for example, // , however,

L.162 , for example,

L. 166 add space between two paragraphs

L.167 am irregular

L.169-170 In Figure 4 the ranges are: 0-3 km, 3-6 km and 6-9 km, while in the text is 0-3 km, 0-6km and 0-9 km. Which is correct? Why did you choose those ranges? If the profile is 10 km, why only 9 km were considered for the CMPs? This needs to be clarified.

L.173 remove "than"

L.171 , whereas many …

L.172 lies

Table 1 change "1000 m" for "10 km" // the offset ranges do not agree with Figure 4

L. 240 , as well as …

L. 244 … refraction, and …//… filter, and …

L.247 again, the offset range do not match with figure 4

L. 251 .. was derived // Is this a range of constant velocities that you have use, or is the final velocity? 5000 – 6500 m/s I will not consider a constant velocity. // … a step range

L.258 New paragraph, then add space between paragraphs

L.262-263 unclear sentence

L. 268 Labeled

L.276 New paragraph, then add space between paragraphs

L.283 summarizes

L.289-292 unclear sentence

L. 294 remove ";"

L.299 why did you use a constant velocity of 5500 m/s for the CDMO, while a range of velocities was used for the DMO and PTSM? This needs to be clarified

L.300 to the west to 40$^o$ to the east with …

L.307 … of the seismic …

L310 in the deeper

L.313 remove "the" before "diffraction"

L.321 to the west

L. 325 Fig. 9a-c

L.326 …, and its coherency decreases (Fig. 9c-f).

Figure 5. Can you add the orientation of the profile? Also, the offset ranges are not the same as Figure 4

L.356357 to address the challenges of the applications of the method in a crystalline rock environment.

L.368 In this study,

Figure 6 same as Figure 5

Figure 7 add the orientation. Also, why are represented 12 km, while only the first 6 km are interpreted. Why chn4-chn6 are not interpreted in this figure?

Figure 8 I cannot see chn4 on Figure 8d-f, neither chn5 in e-f. Am I supposed to see them?. What is chn2 pointing out?

Figure 9, aren't chs3 and chs4 pointing the same reflection? What is chs1 point out?

L.403 remove "the" borfee "regularity#

L.407 with an offset

L.406 Artifacts

L. 407 artifacts

L.412 … of CMPs, especially for longer offsets.

L.413, for example,

L.416 near-surface

L.424 south-dipping

L.439 at the surface

L.444 CDMO towards the west

L.446 structure of

L.449 Unless the north profile was …

L.453 cross-dip elemts

L.470 remove ";"

L.476-480 I do not think this fits in the interpretation section, as it is more like an observation of the results. Can you explain what is related to the diffractions?

L.493 … towards the east or the west.

Figure 10. The diffraction on the shot gather is observed at 0.5 s, while on the stack section is observed at 2.5 km. Can you explain how is diffraction observed so dip, should not be seen around $1.5 - 2$ km? Why do you not compare the shot gather with an unmigrated stacked section (in the time domain)?

L.522 ~8000 ms$^{-1}$

L.525 of

L. 544 remove point before (Mathieu et al., 2020b)

Figure 12. Why the CDP dash line is so short? What does it means? Also, the change on chs2 is observed more towards the right of the dash line. Is it well located?

Section 6.3 can you provided a geological sketch that correlated the north and south interpretations?

L.586 missing spaces "… processing work flow applied in this study …"

L. 596 (Vermeer, 1990, 1998 and 2010)

References

The following citations are missing in the reference list

Cheraghi et al. (2011); Bellefleur et al. (2018); David et al. (2011); Daigneault and Allard (1990); Bedeaux et al. (2020); Vermeer (1994); Dimroth et al., 1995

The following reference is not cited in the text

Juhlin, C.: Finite difference elastic wave propagation in 2D heterogeneous transversely isotropic media, Geophysical Prospecting, 43, no.6, 843–858, 1995b

---

## Author Comment (AC2) · 16 Mar 2021

Fomin Tanya (Referee) tanya.fomin@ga.gov.au General comments: 1. Well written paper with a clear objective and straightforward structure. This case study example is very useful and practical for seismic processors. In these days

PSTM and PSDM methods become most popular and DMO technique called "old fashioned" not used broadly anymore by commercial processing companies (even some seismic software not include DMO in their packages). Reply to comment: First We would like to commend your inclusive review and the detailed comments you provided. We agree that more advanced computing systems have facilitated the application of the sophisticated methods such as PSTM and PSDM in seismic processing, in general. Most of these methods are practiced on sedimentary basins where there is less heterogeneity and scattering. The heterogeneity and scattering is naturally affecting all the seismic surveys acquired in the crystalline rock environment. Yet, the crooked pattern of surveys acquired in crystalline terrain brings more difficulties to seismic processing. We appreciate that you mentioned it is not possible to say which method is the best for crystalline rock environment before it is tested. Our goal was to compare the conventional processing method (DMO stacked migrated) with more advanced method (PSTM), where the survey is crooked, to introduce the challenges. This would help for future seismic survey design in a such environment. DMO/NMO corrected sections followed by stacking and migration is still the most efficient method in crystalline rock environment. For example, the recent seismic surveys acquired in Europe (Smart Exploration program) or 3D seismic surveys in TGI program acquired by Geological survey of Canada, all achieved their best results by standard processing, i.e., DMO corrections/stack/migration. However, we believe that what we learn from current experience would help to better survey design and apply more advanced method in future works. 2. I think one of the difficulties of this topic that you cannot provide "a recipe" what would be the best technique DMO, PSTM or something else for particular geological environment until you test it and apply all possible methods. That is not very practical. It would be good at least if you provide some recommendations on possible processing flows for different geological environments for example DMO should work for some areas and not really useful in others. Reply to comment: We would like to again mention that current experience in crystalline rock terrains recommends that the most applicable method is post-stacked migration processing. In "section 3.1 Offset

distribution for Kirchhoff and DMO correction" and also in "Appendix A. evaluating survey geometry for DMO and PSTM" we emphasized on some major points including regular offset distribution for DMO and PSTM algorithm and also seismic illumination based on subsurface geology. We noted that offset distribution and seismic illumination should be analysed to optimise seismic imaging before acquiring data during the phase of the survey design. The knowledge about the subsurface geology in the study area would improve this analyse. Section 6.1. The effect of survey geometry on seismic imaging, provides some recommendations regarding application of DMO and PSTM algorithm.

3. It is a very detailed interpretation section of seismic reflectivity which is very good but could be over interpreted. Reply to comment: We appreciate that you find the interpretation detailed. In the revised manuscript we provided more concise interpretation. (sections 4, 5, and 6). The interpretation of a regional seismic profile in Chibougamau area including the geological sections or regional models are published somewhere else by some of the co-authors of our paper: Mathieu, L., Snyder, D.B., Bedeaux, P., Cheraghi, S., Lafrance, B., Thurston, P., and Sherlock, R.: Deep into the Chibougamau area, Abitibi Subprovince: structure of a Neoarchean crust revealed by seismic reflection profiling, Tectonics, 38, 1–25, 2020.

Line 20 – methods instead of method Text has been edited (Abstract) Lines 21-22 – What was a reason of 3km increment and was a step 3km or you checked as well 2-4 km, 3-5km etc? Would be 0-3km offset recommendation or it has to be checked for every seismic survey. Reply to comments: The offset step rate of 0-3 km, 3-6 km, and 6-9 km is designed based on the distribution of CMPs for the acquired geometry in Chibougamau area. The offset step rate has to be chosen based on the geometry and could vary for each specific survey (section 3.1 Offset distribution for Kirchhoff PSTM and DMO corrections, the last paragraph) Line 27 – From the Figure 1 it looks like Profile just stops before the Doda fault and not crossing the fault.

The Doda fault is located in the south end of the Chibougamau area beyond the extension of the profile. The surface location of the Doda fault is been updated in Figure 1 based on recent finding of the one of the co-authors of our paper (P. Bedeaux). Section 6.2.2 Seismic interpretation along the south profile, paragraph 5 provides the interpretation about the Doda fault.

Line 54 – Cheraghi et al., 2012 is not on the list of References This Reference is added to the list Line 64 – Bellefleur et al., 2018 is not on the list of References The correct reference is Bellefleur et al. (2019). It is already in the list. Bellefleur et al. (2018) has been changed to Bellefleur et al. (2019) in section 1. Introduction, paragraph 2.

Line 84 – David et al., 2011 is not on the list of References David et al. (2011) has been added to the reference list. Line 103 – Dinmroth et al., 1995 spelling and is not on the list of References Line The proper reference is Dimroth et al. (1985) which is already in the reference list. Dimroth et al. (1985) is cited in section 2. Geological setting, paragraph, 3. 108 – Daigneault and Allard, 1990 is not on the list of References The proper reference is Daigneault et al. (1990) which is already in the reference list and text has been changed to properly cite this reference in section 2. Geological setting, paragraph, 3. Line 111 – Bedeaux et al., 2020 is not on the list of References This reference has been added to the list. Line 127 – How is significant to have more denser VP instead of receiver spacing (cost is more for shots not for channels) Reply to comment: The cost, economic consideration, the logistic and accessibility of the area is considered during survey design to best serve the data acquisition. The geometric consideration of the survey design is published by some of the co-authors of our paper: Naghizadeh, M., Snyder, D.B., Cheraghi, S., Foster, S., Cilensek, S., Feloreani, E., and Mackie, J.: Acquisition and Processing of Wider Bandwidth Seismic Data in Crystalline Crust: Progress with the Metal Earth Project, Minerals, 9 (145), 2019.

What is a Moho depth? Why is only 12 sec record length? Was any testing for higher ending frequencies 150Hz or even higher? The Moho depth is about 36 km ($\sim$ 12 s). A 12 s data is considered to be consistent with the regional survey in the area. The high resolution surveys are processed to image upper crust (0 -12 km). The regional

seismic survey and deeper structures (0- 36 km) in the Chibougamau area is studied in: Mathieu, L., Snyder, D.B., Bedeaux, P., Cheraghi, S., Lafrance, B., Thurston, P., and Sherlock, R.: Deep into the Chibougamau area, Abitibi Subprovince: structure of a Neoarchean crust revealed by seismic reflection profiling, Tectonics, 38, 1–25, 2020. The frequency range is considered based on several pilot tests in the field. The evaluation is explained in a paper published by some of the co-authors of our paper: Naghizadeh, M., Snyder, D.B., Cheraghi, S., Foster, S., Cilensek, S., Feloreani, E., and Mackie, J.: Acquisition and Processing of Wider Bandwidth Seismic Data in Crystalline Crust: Progress with the Metal Earth Project, Minerals, 9 (145), 2019. Line 143 – Common lower case Text has been edited (section 3.1 Offset distribution for Kirchhoff PSTM and DMO corrections, paragraph 1). Line 169- 170 – "We designed offset . . . " Was this designed only based on visual assessment or something else? Reply to comment: It has been explained in section 3.1 Offset distribution for Kirchhoff PSTM and DMO corrections, paragraph 4: These offset ranges are chosen based on analysis shown in Fig. 2 and Fig.3 and testing the seismic images of variety of offset ranges when they contribute to the process of post-stacked DMO and PSTM images (see Table 2 for the processing details). The offsets greater than 9 km did not increased the image quality and deemed unnecessary to present their images.

Line 221 – You don't need to have "The distribution. . ." sentence second time. Text has been edited (caption for Figure 4). In the Figure 4 offsets 0-3km, 3-6km and 6-9km. In the text, it is 0-3km, 0-6km and 0-9km. Am I wrong of reading that? Reply to comment: In "section 3.1 Offset distribution for Kirchhoff PSTM and DMO corrections, paragraph 4" we explain that: We designed offset planes ranging 0-3 km, 0-6 km, and 0-9 km in order to study the survey geometry (Fig. 4). These offset ranges are chosen based on analysis shown in Fig. 2 and Fig.3 and testing the seismic images of variety of offset ranges when they contribute to the process of post-stacked DMO and PSTM images (see Table 2 for the processing details). The offsets greater than 9 km did not increased the image quality and deemed unnecessary to present their images. All seismic images in Figure 5 and 6 are generated in offset range of 0-3 km, 0-6 km, and

0-9 km and it has been mentioned in text. In Figure 4 instead of overprinting of CMP distribution for offset range of 0-3 km, 0-6 km, and 0-9 km, we show CMPs of offset range of 0-3 km, 3-6 km, and 6-9 km, to present their distribution along the survey line where offset is increasing.

In Table 2 First arrivals picked up to 10 km. Is any need for that? I assume this is a one-layer refractor model? Why top muting but not just stretch with some %? Reply to comment: Top mute would help to remove first arrivals in short and longer offsets and prevents the removing the potential reflections in shallower parts. Line 251 – Why was used a constant velocity for DMO corrections? Reply to comment: The constant velocity model for DMO corrections is based on several test including constant and variable velocities. The velocity of 5500 m/s showed the best results.

Figures 5 and 6 should be Depth converted migrated sections? Both sections are time-to-depth converted after migration (Table 2).

Line 385 – See capital Text has been edited (caption for Figure 7). Figure 9 You need better arrow for the fault location (similar to figure 8) Figure 9 has been updated regarding this comment. Line 518 - shot gather 15135 but in the figure 13135 Text has been edited (Caption for Figure 12) Line 648 – Vermeer, 1994 is not on the list This reference has been added to the reference list. Line 652 – 653 " The pre-stack depth migration . . ." something missing in this sentence? Text has been edited (8 Appendix A: evaluating survey geometry for DMO and PSTM, the last paragraph) Line 697 – Is it 2018? Line 738 – I could not find this reference in the text. The correct reference is Bellefleur et al. (2019) which is already in the reference list. Bellefleur et al. (2018) has been changed to Bellefleur et al. (2019) in section 1. Introduction, paragraph 2. Final remarks It is a good and useful paper for people who process seismic data particular for hard rock data sets. We need to be very careful and not to over interpret reflection seismic data by trying to fit to geological model. Reply to comment: One again, we would like to acknowledge Mrs. Fomin for her inclusive comments which improved the quality our paper. We would like to clarify that the inclusive interpretation of seismic sections such

as fault kinematic, structural and tectonic study is beyond the scope of our study and it is published somewhere else: Mathieu, L., Snyder, D.B., Bedeaux, P., Cheraghi, S., Lafrance, B., Thurston, P., and Sherlock, R.: Deep into the Chibougamau area, Abitibi Subprovince: structure of a Neoarchean crust revealed by seismic reflection profiling, Tectonics, 38, 1–25, 2020. Some of the co-authors of our paper were contributed in this publication and helped to improve the interpretation of the high resolution seismic sections in the Chibougamau area.

---

## Author Comment (AC4) · 16 Mar 2021

**Reviewer #2**

Comments on Seismic imaging across fault systems in the Abitibi greenstone belt – An analysis of pre- and post-stack migration approaches in the Chibougamau area, Quebec, Canada, by Saeid Cheraghi et al

This paper discusses the processing strategy to apply to vertical incidence seismic reflection data in order to get best subsurface image along crooked-line acquisitions. In that regard, I think the paper is excellent. The analysis it makes of the importance of CDP bin centers in relation to CDPs location and maximum offset, together with the need of undertaking cross-dip move out tests is crucial to get the best resolved image. However, I'm a bit disappointed about the interpretation of the data. I don't know if the goal of this paper is to present also a geological model of the area, as they don't really do it. At this point, I don't see a clear relationship between the faults and the reflectivity. As an example, the Doda fault does not seem to have a seismic response in the southern profile. The width of the reflectivity associated to the Guercheville and Barlow faults is much higher than the trace of the faults themselves. So what is the reflectivity responding to? Deformation or lithological contrast? Is the later related to Au mineralization or not? In relation to this, the map in figure 1 lacks information about dips and a cross-section where we can have an idea of the structure. Finally, to round up the conclusions, both profiles should be plotted overlapping the entire seismic profile, presenting the overall structure of the area. As it is now, the paper is a good technical work with limitations regarding the interpretation. Finally, even though there are native English speaking researches among the authors, I find the text awkward sometimes. Conclusions read like a telegram, and some parts of the text do not flow properly. So a revision of the grammar and style would be convenient (from my point of view).

*Reply to comment:*

First, we would like to appreciate the reviewer for their inclusive comments and mentioning the quality of our paper excellent. It is really encouraging us. As we mentioned in text the main concept of our paper is to address challenging associated with acquiring crooked seismic profiles in crystalline rock terrains. We mainly explained our research to optimize DMO and PSTM process. We also explained the interpretation of the seismic sections to provide more insight about the effect of the processing methods. Detailed interpretations about the kinematic of faults, providing geological sections or study the structures in regional scale is beyond the scope or study. The regional seismic section in Chibougamau area and its interpretations were recently published elsewhere by some of the co-authors of this paper and these are not the focus of this paper. We have cited this publication in our paper:

Mathieu, L., Snyder, D.B., Bedeaux, P., Cheraghi, S., Lafrance, B., Thurston, P., and Sherlock, R.: Deep into the Chibougamau area, Abitibi Subprovince: structure of a Neoarchean crust revealed by seismic reflection profiling, Tectonics, 38, 1–25, 2020.

Again, we would like to mention that the kinematic study of the faults needs field measurements and it is not the focus of our research here. We would like to mention that the kinematic of the Barlow fault is published by one of the co-authors of this paper:

Bedeaux, P., Brochu, A., Mathieu, L., Gaboury, D., Daigneault, R.: Structural analysis and metamorphism of the Barlow Fault Zone, Chibougamau area, Neoarchean Abitibi Subprovince: Implications for gold mineralization, Canadian Journal of earth Sciences, accepted, 2020.

Now, in section 6 we have provided more details about the interpretation of the structures and specially the Barlow fault, the Guercheville fault, and the Doda fault. Potential area for mineral exploration (orogenic gold) is also discussed in section 6.

Figure 1 has changed to present axis of the major anticlines and synclines in the Chibougamau area which provides a better taste about dip and strike of the regional structures. The legend in Figure 1 now presents a stratigraphic column with age of the major formations. The strike of the Doda fault on Figure 1 is updated based on finding of recent field work study by one of the co-authors (P. Bedeaux).

We believe that while the goals of regional and high resolution seismic studies are different there is no need to republish the regional image here.

Line 16-17: Would help to know what type of faults are them. Strike slip, normal or reverse? As there is not a proper stratigraphic column in the legend, the kinematics of the faults is difficult to infer

*Reply to comment:*

We mentioned that the kinematic studies are not the scope of our research (last paragraph of the introduction). How ever, more information about the Barlow fault is added to text based on recent field studies of P. Bedeaux (section 2. Geological setting, paragraph 3).

Line 23: In the northern?

*Reply to comment:*

text is edited (Abstract).

Line 24: structures or the key geological structure….

*Reply to comment:*

Text is edited (Abstract)

Line 83: Fig. 1 does not show …the NE portion of the Abitibi super-province. A regional scale map (at least as the inset in figure 1) where this province appeared will be much better.

*Reply to comment:*

Figure 1 has been edited regrading this comment. The inset on Figure 1 shows the Abitibi subprovince and the study area. The Canada map also shows the location od the study area.

Line 84: Same applies for the ages of rocks. Include them in the legend of Figure 1 and not only in the text so we can have a quick idea of the structure. Right now we don't know which ones are older or younger.

*Reply to comment:*

Now, the legend of Figure 1 shows the stratigraphic column and ages of the rocks.

Line 252: …various constant velocities between 5000-6500 m/s, with a step range…..?

*Reply to comment:*

Text has been edited (section 4. Dara processing and results, paragraph 2).

Line 258. Here you start presenting results but the whole thing is quite messy. A new paragraph (add inter-paragraph space as you do in other places, e.g., between lines 295 and 296) should start in line 258 and another one in line 264 (The design of the north….)

*Reply to comment:*

Text has been edited regarding this comment (section 4. Dara processing and results, paragraph 3 and 4).

Line 268: ….Labelled in Fig. 5, chn1……….

*Reply to comment:*

Text has been edited (section 4. Dara processing and results, paragraph 4).

Line 271. Those reflections project to the surface off the CDP line, so rephrase. Is not that they show no correlation with the surface geology. We don't see it. And the map you provide in Fig 1 lacks detail, but some could be related with the Burlow pluton southern boundary?

*Reply to comment:*

Text has been edited regarding this comment. We now interpret them as they could be related to southern boundary of the Barlow pluton (section 4. Dara processing and results, paragraph 4).

Line 282: I don't see chs5 and6 as subhorizontal. If something, chs5 has a hyperbolic geometry with high opposite dip to ch6 in its deeper part

*Reply to comment:*

Text has been edited and now we mention them as steeply dipping reflections (section 4. Dara processing and results, paragraph 5).

Line 272:….one kilometer…you are using km and numbers so it should be 1 km.

*Reply to comment:*

Text has been edited (section 4. Dara processing and results, paragraph 4).

Line 276: New paragraph again (interparagraph space)

*Reply to comment:*

Text has been edited (section 4. Dara processing and results, paragraph 5).

Line 284: I'd like that title to be more specific.

*Reply to comment:*

Text has been edited. Now it has been changed to "section 5. cross-dip analysis".

Line 289: Change to……When out-of-place CMP's scattered/reflected seismic waves from steep structures off the CDP line (cross-dip direction) exist, cross-dip analysis addresses…..??As it is now, the phrase does not read well. I think you make an excesive use of semi-colon when you could replace it by commas or new paragraphs.

*Reply to comment:*

Text has been edited to follow this comment (section 5. Cross-dip analysis paragraph 1).

Line 311: Remove…Table 3 shows…..segment. It is already mentioned in the previous paragraph

*Reply to comment:*

It has been removed from text (section 5. Cross-dip analysis paragraph 3).

Line 312: Remove…Table 3 shows…..segment. It is already mentioned in the previous paragraph.

*Reply to comment:*

It has been removed from text (section 5. Cross-dip analysis paragraph 3).

Line 314:……depths lesser than…..

*Reply to comment:*

Text has been changed (section 5. Cross-dip analysis paragraph 3).

Line 322: …40º to the south and features lesser continuity (Fig. 9c). You should also take into account the continuity of the reflection

*Reply to comment:*

Text has been changed (section 5. Cross-dip analysis paragraph 4).

Line 356: High resolution seismic profiles….

*Reply to comment:*

Text has been changed (section 6. Discussion)

Lines 435: Unconformities are identified in vertical incidence data when the reflections they truncate are visible???. In my opinion it is more likely that chn1 responds to lithological variations inside the Opemisca Group.

*Reply to comment:*

Text has been changed (section 6.2.1. Seismic interpretation along the north profile, paragraph 2): "correspond to internal structure such as an unconformity or small fault that is part of the Waconichi Tectonic Zone or lithological variations inside the Opémisca Group".

Lines 436: This interpretation of chn2 is incomplete. What is the structure of the Opemisca Group there? Why doesn't it respond to another change in lithology?

*Reply to comment:*

Text has been changed (section 6.2.1. Seismic interpretation along the north profile, paragraph 2):

Text has been changed: "Similar to reflection chn1, Reflection chn2 (Fig. 5, Table 3) correlates with local structure, i.e., small fault or mafic/ultramafic lithology in outcrops of Opémisca Group rocks.

Line 443 and fw: Rewrite, as there are articles missing. For instance:

The CDMO analysis around reflections chn3 (Fig. 8) would suggest a 0°-10° strike towards the east (Fig. 8c and 8d, Table 3). Furthermore, these reflections became weakly imaged assuming a CDMO toward the west (Fig. 8a 445 and 8b) or toward the east at dips greater than 10° (Fig. 8e and 8f). Finally, the CDMO analysis also indicates an eastward apparent dip for other upper crustal reflection packages of the north profile (chn1 and chn2, Table 3).

*Reply to comment:*

Text has been changed to follow this comment (section 6.2.1. Seismic interpretation along the north profile, paragraph 3).

Line 456:………provided insights….

*Reply to comment:*

Text has been changed (section 6.2.1. Seismic interpretation along the north profile, paragraph 5)

Linbe 457:…..they are potentially relevant….

*Reply to comment:*

Text has been changed (section 6.2.1. Seismic interpretation along the north profile, paragraph 5)

Line 456-480: You need to rewrite that part as you merely do a description, but not a discussion. So discuss, e.g., why CDMO should help in imaging diffractions given the geometry of the waves in the latter. Also, you can discuss what you think they represent. As it is now, there is no discussion in there.

*Reply to comment:*

Text has been changed to follow this comment (section 6.2.1. Seismic interpretation along the north profile, paragraphs 5-9).

Line 524-525: This is a really interesting problem. But there should be ways to address it. The south profile is oblique to the fault at the cross-point so in Figure 9c you are imaging an apparent dip. But apart from knowing the angle between the profile and the fault at that point, you also know that real dips are higher than apparent dips. So probably the best image of the fault is that where its reflection looks shorter and steeper. This should help you to provide the real geometry associated to that feature. Moreover, addressing your preferred image of the fault in figure 9 is part of what a discussion should be, instead of just saying that the fault has a complex geometry (something that is not clear from the map, as it looks subvertical and simple).

*Reply to comment:*

Text has been changed to follow this comment (6.2.2 Seismic interpretation along the south profile

, paragraphs 2-3).

.

Lines 531: What is the dip of rocks at the surface??? This implies again the need of a geological map with layer dips and a cross-section.

*Reply to comment:*

Now, fold axis for all the folds known in the study area are shown in Figure 1

Line 536: If interpreted as a fault, reflection chs4 most likely correlates to the Doda fault?. But Doda fault projects at CDP 100 and these reflections project outside the profile! Even if dip changes and the fault becomes subvertical, you are imaging things at 2 km in the migrated section and nothing at the surface where the Doda Fault projects. It seems unlikely that chs4 represents that fault. Finally, at the cross-point with the Doda Fault, the profile is oblique, so you are seeing apparent dips. Does Figure 9 provide better insights about this fault? This should be better discussed.

*Reply to comment:*

Text has been changed regarding this comment (6.2.2 Seismic interpretation along the south profile

, paragraphs 5).

Line 541: Those reflections are not subhorizontal. They have opposite dips and suggest a syncline structure.

*Reply to comment:*

Text has been edited (6.2.2 Seismic interpretation along the south profile, paragraphs 6).

Lines 557-558:…. joint complex structure of the Guercheville fault as well as the Doda fault in the south are all imaged within the greenstone belt rocks of the upper….

*Reply to comment:*

Text has been edited (section 6.3 Potential for exploration of orogenic gold).

Line 559:…deep reflections: chn5 and chn6…..Do not mix reflectors and reflections.

*Reply to comment:*

Text has been edited (section 6.3 Potential for exploration of orogenic gold).

Conclusions: I don't see the Doda Fault anywhere in the seismic section. It projects around CDP 100 in the seismic profile and there, the strongest reflectivity is at 2-3 km. You need to discuss that in the discussion part, but it is not a clear conclusion with the present discussion.

*Reply to comment:*

Text has been changed regarding this comment. As we explained above the strike of the Doda fault has been edited on the map (Figure 1). We discussed the association of reflection chs4 to the Doda fault in text (6.2.2 Seismic interpretation along the south profile, paragraphs 5).

Table 2: Is it necessary to be that specific in step 6? Is not enough to say V2t?

Table 2 has been edited regarding this comment.

Does the time to depth conversion use velocities higher than migration velocities? Explain or change

*Reply to comment:*

Our bad to forget update Table 2 regarding our processing flow. The migration velocity is 5500 at surface (0 s) (based on first arrival velocities) and 6200 at 4 s (crustal velocity). Table 2 has been edited. Figures 5-6 are updated regarding migration velocity and then the sections are time-to-depth converted.

Figure 1: Geological map should have dips. Furthermore, a cross section along the profile should be presented. I'm sure there are has been plenty of structural geologists working on that.

*Reply to comment:*

Figure 1 has been edited to present axis of major anticlines and synclines. Presenting the geological sections is beyond the scope of our study. We explained above that some of the co-authors of our paper have been published articles recently which includes geological sections, tectonic studies, kinematic of faults.

Figure 4: Caption-…………..shot and CDP locations are also……

Caption has been edited.

Figure 5, 6 etc: Add N and S in the edges of the profile. Although the reader can figure out the dip of reflections, it is faster to indicate the orientation in the profile itself.

Figures are updated to present N and S

---

## Author Comment (AC5) · 16 Mar 2021

**Reviewer 3**

The authors of the manuscript titled "Seismic imaging across fault systems in the Abitibi greenstone belt – An analysis of pre- and post-stack migration approaches in the Chibougamau area, Quebec, Canada" presents a processing strategy to image strong dipping reflections from a crooked-line acquisition. The manuscript is well written and the methodology is well described. Although, the geological interpretation is not really developed. The manuscript should be a significant and valuable work, and it fits the scope of Solid Earth's special issue "State of the art in mineral exploration". However, some suggestions and technical comments are as follow:

*Reply to comment:*

We would like to appreciate the reviewer for their positive opinion about our research. We have explained in the introduction of our paper which the main goal of our research is to address optimizing application of the DMO and PSTM in crystalline rock environment where majority of the seismic profiles are crooked. The inclusive interpretation including Kinematic of faults or structural studies are not our main goal. However, we provide more details about the interpretation of the seismic profiles in the revised version in section 4 data processing and results, and section 6 discussion. We have cited publications from some of the co-authors of our paper in Chibougamau are with focused view on geological interpretation:

Mathieu, L., Snyder, D.B., Bedeaux, P., Cheraghi, S., Lafrance, B., Thurston, P., and Sherlock, R.: Deep into the Chibougamau area, Abitibi Subprovince: structure of a Neoarchean crust revealed by seismic reflection profiling, Tectonics, 38, 1–25, 2020.

Bedeaux, P., Brochu, A., Mathieu, L., Gaboury, D., Daigneault, R.: Structural analysis and metamorphism of the Barlow Fault Zone, Chibougamau area, Neoarchean Abitibi Subprovince: Implications for gold mineralization, Canadian Journal of earth Sciences, accepted, 2020.

1) It is not clear why the different offset ranges of 0-3, 3-6, and 6-9 km are selected, even considering that, the seismic profile is 10 km. In addition, during the text, these ranges are changing from 0-3, 3-6, and 6-9 km to 0-3, 0-6 and 0-9 km. Which is the correct one?

*Reply to comment:*

In "section 3.1 Offset distribution for Kirchhoff PSTM and DMO corrections, paragraph 4" we explain that: We designed offset planes ranging 0-3 km, 0-6 km, and 0-9 km in order to study the survey geometry (Fig. 4). These offset ranges are chosen based on analysis shown in Fig. 2 and Fig.3 and testing the seismic images of variety of offset ranges when they contribute to the process of post-stacked DMO and PSTM images (see Table 2 for the processing details). The offsets greater than 9 km did not increased the image quality and deemed unnecessary to present their images.

All seismic images in Figure 5 and 6 are generated in offset range of 0-3 km, 0-6 km, and 0-9 km and it has been mentioned in text. In Figure 4 instead of overprinting of CMP distribution for offset range of 0-3 km, 0-6 km, and 0-9 km, we show CMPs of offset range of 0-3 km, 3-6 km, and 6-9 km, to present their distribution along the survey line where offset is increasing.

2) In the interpretation, some of the reflections are associated with different geological structures. However, the geological map does not provide any strike and dip information. Also, the profile, to

the south, does not cross the Doda fault while in the interpretation one of the reflections is associated with the fault. In the conclusions, it is mentioned that this fault is only imaged in the first 2 km, while the chs4 (associated with the fault) is observed around 2–3 km depth. Are you sure that chs4 is the fault? In the text, it is not clear the origin of the diffractions. Is it the fault or a potential ore body?

*Reply to comment:*

Figure 1 has been changed to show major axis of the folds (anticline/syncline) in the Chibougamau area. Also, the strike of the Doda fault has been edited in Figure 1 based on recent finding of one of the co-authors in our paper (P. Bedeaux). In "section 6.2.2 Seismic interpretation along the south profile, paragraph 5" we interpret chs4 and its possible association with the Doda fault. In section 6.2.2 we explain that the Doda fault is measures subvertical in the field and chs4 may image this fault at depths greater than 2 km. This is consistent with what we say in the conclusion.

The diffractions are interpreted potentially as orebodies (section 6.2.1 Seismic interpretation along the north profile, paragraphs, 5-9)

3) It is possible to provide a geological model of the final interpretation? Which are the relationship between chn1 and chs1?

*Reply to comment:*

The distance between the areas associated to chs1 and chn1 is ~ 50 km on the map (Figure 1). Providing a regional model/interpretation is beyond the scope of our study. The interpretation of the regional seismic profile in Chibougamau area including the geological sections and regional model are published somewhere else by some of the co-authors of our paper:

Mathieu, L., Snyder, D.B., Bedeaux, P., Cheraghi, S., Lafrance, B., Thurston, P., and Sherlock, R.: Deep into the Chibougamau area, Abitibi Subprovince: structure of a Neoarchean crust revealed by seismic reflection profiling, Tectonics, 38, 1–25, 2020.

4) Can you provide the orientation of the seismic profiles in Figures 5–12? In Figure 7, why is represented 12 km, if only there are interpreted the first 6 km, why chn4-chn6 are not to interpreted in this figure? Also, in Figure 10, the chn_diff seems not agree on the shot gather and the stacked section.

*Reply to comment:*

Now, the orientation of the profiles is shown in figures. Entire depth of the section i.e., 4 s (12 km) is shown in Figure 7 first to show the effect of CDMO on both shallow and deeper part and observe if CDMO can enhance the coherency of the reflections in deeper part and second to be consistent with other sections shown in text. They all present depths between 0-4 s (0-12 km).

Reflection sets chn4, chn5, and chn6 are better imaged in the second segment of the profile shown in Figure 8. The interpretation is provided based on imaging properties in Figure 8 (6.2 Seismic interpretation in Chibougamau area)

Figures 7-11 are update to show time for depth variations of the CDMO stacked sections. Table 3 shows that the velocity of 6000 m/s is considered for time-to-depth conversion after migration shown in Figures

5-6. Figure 10a and Figure 8c is referring to the shallow diffraction (~ 2.5 km/0.75 s) imaged in reflection package chn3 marked with dashed ellipse in Figure 8.c. Figure 10 b show the location of this diffraction at ~ 0.75 s which is not at the same depth/location diffraction chn_diff is imaged. Diffraction chn_diff is shown in Figure 8b-c (at depth of ~ 4 km/1.5 s). Figure 11 shows its location in a shot gather at ~ 1.5 s.

5) In the text is mentioned that for DMO it is used a range velocity of 5000–6500 m/s, while for the CDMO is used a constant velocity of 5500 m/s. Why this decision was made, can you explain it further? In addition, in my understanding, a range of velocities, such as 5000–6500 m/s, will not be considered a constant velocity.

*Reply to comment:*

Cross dip moveout is applied to DMO corrected sections with constant velocity of 5500 m/s. Velocities between 5000-6500 m/s are utilized for stacking after CDMO (section 5. Cross-dip analysis, paragraph 2).

6) Some references are missing.

It has been checked. Now all the references are properly cited and listed.

Minor comments:

L.15 Canada,

Text has been (Abstract)

L.17-18 with a known metal endowment in the area

Text has been edited (Abstract)

L.24 key geological structures // sets

Text has been edited (Abstract)

Introduction, paragraph 1

L.35-36 (Juhlin, 1995a; Juhlin et al., 1995 and 2010; Bellefleur et al., 1998 and 2015; Perron and Calvert, 1998; Ahmadi et al., 2013)

Text has been edited (section 1. Introduction, paragraph 1)

L. 36 add comma after "however"

Text has been edited (section 1. Introduction, paragraph 1)

L. 67 Mercier-Langevin et al., 2014

Text has been edited (section 1. Introduction, paragraph 1)

L.33 crystalline rock

Text has been edited (section 1. Introduction, paragraph 1)

L. 35-36 Juhlin, 1995a; Juhlin et al., 1995, 2010; Bellefleur et al., 1998, 2015 … // remove ; // . However, all …

Text has been edited (section 1. Introduction, paragraph 1)

L.37 than a coherent

Text has been edited (section 1. Introduction, paragraph 1)

L. 38 Petrophysical measurements, where available, complemented with reflectivity/velocity models of the shallow crust, …

Text has been edited (section 1. Introduction, paragraph 1)

L. 39 permit a more

Text has been edited (section 1. Introduction, paragraph 1)

L.42 limited the application

Text has been edited (section 1. Introduction, paragraph 1)

L.45-46 confusing sentence. Can you rephrase it?

Text has been edited (section 1. Introduction, paragraph 1)

L.47 change "which" for "that"

Text has been edited (section 1. Introduction, paragraph 1)

L.59 Bedides

Text has been edited. Beside is replaced with besides (section 1. Introduction, paragraph 2)

L.61-63 confusing sentence. Can you rephrase it?

Text has been edited (1. Introduction, paragraph 2)

L.78 Are the offset ranges between 0-10 km or 0-9 km? In the abstract is mentioned 0-9 km, while here is between 0 and 10 km. It is somewhat confusing.

Section 1. Introduction, paragraph 3: it has been edit to show 0-9 km.

L.90felsic lava flows

Text has been edited (Section 2. Geological setting, paragraph 1)

L. 92 Mueller et al., 1989; Leclerc et al., 2017

Text has been edited (Section 2. Geological setting, paragraph 1)

L.92-93 …, observed along the southern profile, …

Text has been edited (Section 2. Geological setting, paragraph 1)

L.102 , and

Text has been edited (section 2. Geological setting, paragraph 2)

L. 103 Dimroth et al., 1995

Text has been edited (section 2. Geological setting, paragraph 2)

L.106 change "to" for "with"

Text has been edited (section 2. Geological setting, paragraph3)

L. 108-109 Leclerc et al., 2012 and 2017 // schistosity

Text has been edited (section 2. Geological setting, paragraph3)

L.114 at the surface

Text has been edited (section 2. Geological setting, paragraph3)

L. 117 Dimroth, 1985; Mueller et al., 1989

Text has been edited (section 2. Geological setting, paragraph3)

L.120-121 confusing sentence. Can you rephrase it?

Text has been edited (Section 3. Seismic data section, paragraph 1)

L. 148 low-velocity

we did not see this in text to edit.

L.142 DMO or PSTM,

Text has been edited (section 3.1. Offset distribution for Kirchhoff PSTM and DMO corrections, paragraph 1)

L.152 artifacts

Text has been edited (section 3.1. Offset distribution for Kirchhoff PSTM and DMO corrections, paragraph 2)

L.158 . for example,

Text has been edited (section 3.1. Offset distribution for Kirchhoff PSTM and DMO corrections, paragraph 3)

L.161 , for example, // , however,

Text has been edited (section 3.1. Offset distribution for Kirchhoff PSTM and DMO corrections, paragraph 3)

L.162 , for example,

Text has been edited (section 3.1. Offset distribution for Kirchhoff PSTM and DMO corrections, paragraph 3)

L. 166 add space between two paragraphs (the space is added)

Text has been edited (section 3.1. Offset distribution for Kirchhoff PSTM and DMO corrections, paragraph 3)

L.167 am irregular (we assume the reviewer wants to say an irregular)

Text has been edited (section 3.1. Offset distribution for Kirchhoff PSTM and DMO corrections, paragraph 4)

L.169-170 In Figure 4 the ranges are: 0-3 km, 3-6 km and 6-9 km, while in the text is 0-3 km, 0-6km and 0-9 km. Which is correct? Why did you choose those ranges? If the profile is 10 km, why only 9 km were considered for the CMPs? This needs to be clarified.

*Reply to comment:*

In "section 3.1 Offset distribution for Kirchhoff PSTM and DMO corrections, paragraph 4" we explain that: We designed offset planes ranging 0-3 km, 0-6 km, and 0-9 km in order to study the survey geometry (Fig. 4). These offset ranges are chosen based on analysis shown in Fig. 2 and Fig.3 and testing the seismic images of variety of offset ranges when they contribute to the process of post-stacked DMO and PSTM images (see Table 2 for the processing details). The offsets greater than 9 km did not increased the image quality and deemed unnecessary to present their images.

All seismic images in Figure 5 and 6 are generated in offset range of 0-3 km, 0-6 km, and 0-9 km and it has been mentioned in text. Only in Figure 4 we show offset range of 0-3 km, 3-6 km, and 6-9 km, to present CMP distribution along the survey line where offset is increasing.

L.173 remove "than"

Text has been edited (section 3.1. Offset distribution for Kirchhoff PSTM and DMO corrections, paragraph 4)

L.171 , whereas many …

Text has been edited (section 3.1. Offset distribution for Kirchhoff PSTM and DMO corrections, paragraph 4)

L.172 lies

Text has been edited (section 3.1. Offset distribution for Kirchhoff PSTM and DMO corrections, paragraph 4)

Table 1 change "1000 m" for "10 km" // the offset ranges do not agree with Figure 4

*Reply to comment:*

Table 1 does not present offsets. Table 2 shows the offset range. Text is edited to show 10 km. Please see above our reply about Figure 4 and the offsets ranges utilized in our processing.

L. 240 , as well as …

Text has been edited (section 4. Data processing and results, paragraph 1)

L. 244 … refraction, and …//… filter, and …

Text has been edited (section 4. Data processing and results, paragraph 1)

L.247 again, the offset range do not match with figure 4

*Reply to comment:*

Please see our reply about Figure 4 above mentioned

L. 251 .. was derived // Is this a range of constant velocities that you have use, or is the final velocity? 5000 – 6500 m/s I will not consider a constant velocity. // … a step range

Text has been edited (Section 4. Data processing and results, paragraph 2): "This chosen velocity derived from several tests using various constant velocities between, 5000-6500 m/s, with step range of 100 m/s".

L.258 New paragraph, then add space between paragraphs

Text has been edited (section 4. Data processing and results, paragraph 3)

L.262-263 unclear sentence

Text has been edited (section 4. Data processing and results, paragraph 3)

L. 268 Labeled

Text has been edited (section 4. Data processing and results, paragraph 3)

L.276 New paragraph, then add space between paragraphs

Text has been edited regarding this comment (Text has been edited (section 4. Data processing and results, paragraph 5).

L.283 summarizes

Text has been edited (section 4. Data processing and results, paragraph 5)

L.289-292 unclear sentence

Text has been edited (section 5. Cross-dip analysis, paragraph 1)

L. 294 remove ";"

Text has been edited (section 5. Cross-dip analysis, paragraph 1)

L.299 why did you use a constant velocity of 5500 m/s for the CDMO, while a range of velocities was used for the DMO and PTSM? This needs to be clarified

*Reply to comment:*

DMO corrected section with constant velocity of 5500 m/s is considered for CDMO. Then, time delays of CDMO is calculated and applied. Finally, we stacked DMO-CDMO corrected traces using a velocity model designed from the one applied after DMO corrections during standard processing (Table 2). This is mentioned in section 5. Cross-dip analysis, paragraph 2

L.300 to the west to 40⁰ to the east with …

Text has been edited (section 5. Cross-dip analysis, paragraph 2)

L.307 ... of the seismic …

Text has been edited (section 5. Cross-dip analysis, paragraph 3)

L310 in the deeper

Text has been edited (section 5. Cross-dip analysis, paragraph 3)

L.313 remove "the" before "diffraction"

Text has been edited (section 5. Cross-dip analysis, paragraph 3)

L.321 to the west

Text has been edited (section 5. Cross-dip analysis, paragraph 4)

L. 325 Fig. 9a-c

Text has been edited (section 5. Cross-dip analysis, paragraph 4)

L.326 …, and its coherency decreases (Fig. 9c-f).

Text has been edited (section 5. Cross-dip analysis, paragraph 4)

Figure 5. Can you add the orientation of the profile? Also, the offset ranges are not the same as Figure 4

*Reply to comment:*

The orientation of the line is added now to this figure. Please see our comment about Figure 4 mentioned above

L.356357 to address the challenges of the applications of the method in a crystalline rock environment.

Text has been edited (section 6. Discussion, paragraph 1)

L.368 In this study,

Text has been edited (section 6.1 The effect of survey geometry on seismic imaging, paragraph 1)

Figure 6 same as Figure 5

The labels have ben added to show the orientation of the profile.

Figure 7 add the orientation. Also, why are represented 12 km, while only the first 6 km are interpreted. Why chn4-chn6 are not interpreted in this figure?

*Reply to comment:*

Figure 7 is updated to show time for depth variations. Entire depth of the section i.e., 12 km (4s) is shown in Figure 7 first to show the effect of CDMO on both shallow and deeper part and observe if CDMO can enhance the coherency of the reflections in deeper part and second to be consistent with other sections shown in text. They all present depths between 0-4 s (0-12 km).

Reflection sets chn4, chn5, and chn6 are better imaged in the second segment of the profile shown in Figure 8. The interpretation is provided based on imaging properties in Figure 8 (6.2 Seismic interpretation in Chibougamau area)

Figure 8 I cannot see chn4 on Figure 8d-f, neither chn5 in e-f. Am I supposed to see them?. What is chn2 pointing out?

*Reply to comment:*

Figure 8 is updated to show time for depth variations. Figure 8 present the application of the CDMO. The section without CDMO is shown in Figure 8c and all the reflections marked and interpreted in Figure 5 are shown here. Then, after application of CDMO to the west or to the east some of the reflections show more coherency some less. The location of the all reflections are shown for better comparison/evaluation. Interpretation of reflection chn2 is shown in section 6.2.1 Seismic interpretation along the north profile, paragraph 2: "Similar to reflection chn1, Reflection chn2 (Fig. 5, Table 3) correlates with local structure, i.e., small fault or mafic/ultramafic lithology in outcrops of Opémisca Group rocks".

Figure 9, aren't chs3 and chs4 pointing the same reflection? What is chs1 point out?

Reflection sets chs3 and chs4 are two separate packages. See section 6.2.2. Seismic interpretation along the south profile, paragraph 4-5 for interpretation.

For interpretation of reflection chs1 see section 6.2.2. Seismic interpretation along the south profile, paragraph 2.

L.403 remove "the" borfee "regularity#

Text has been edited (section 6.1. The effect of survey geometry on seismic imaging, paragraph 2)

L.407 with an offset

Text has been edited (section 6.1. The effect of survey geometry on seismic imaging, paragraph 2)

L.406 Artifacts

Text has been edited (section 6.1. The effect of survey geometry on seismic imaging, paragraph 2)

L. 407 artifacts

Text has been edited (section 6.1. The effect of survey geometry on seismic imaging, paragraph 2)

L.412 … of CMPs, especially for longer offsets.

Text has been edited (section 6.1. The effect of survey geometry on seismic imaging, paragraph 2)

L.413, for example,

Text has been edited (section 6.1. The effect of survey geometry on seismic imaging, paragraph 2)

L.416 near-surface

Text has been edited (section 6.2. Seismic interpretation in Chibougamau area)

L.424 south-dipping

Text has been edited (section 6.2.1. Seismic interpretation along the north profile, paragraph 1)

L.439 at the surface

Text has been edited (section 6.2.1. Seismic interpretation along the north profile, paragraph 3)

L.444 CDMO towards the west

Text has been edited (section 6.2.1. Seismic interpretation along the north profile, paragraph 3)

L.446 structure of

Text has been edited (section 6.2.1. Seismic interpretation along the north profile, paragraph 3):

"originates within a complex structure, off the plane of the north profile".

L.449 Unless the north profile was …

Text has been edited (section 6.2.1. Seismic interpretation along the north profile, paragraph 4):

L.453 cross-dip elemts

Text has been edited (section 6.2.1. Seismic interpretation along the north profile, paragraph 4)

L.470 remove ";"

Text has been edited (section 6.2.1. Seismic interpretation along the north profile, the last paragraph)

L.476-480 I do not think this fits in the interpretation section, as it is more like an observation of the results. Can you explain what is related to the diffractions?

*Reply to comment:*

Diffraction are interesting objects in exploration seismology as they could refer to small lenses which in mineral exploration could be an orebody. Diffraction imaging is always a challenging fact in seismology and the observed diffraction needs to be supported in stacked sections (depth converted) as well as the shot gathers (time domain). In section 5 Cross-dip analysis we just comment about diffraction imaging. In section 6.2.1. Seismic interpretation along the north profile, paragraphs 5-9 we inclusively addressing the diffraction imaging/validity of imaging, its potential for mineral exploration and also justify our results when we show the diffractions in shot gathers. There are two diffractions along the north profile (See Figure 8). In section 6.2.1. Seismic interpretation along the north profile, we discuss that CDMO can be applied for better observation of the diffractions.

L.493 ... towards the east or the west.

Text has been edited (section 6.2.2. Seismic interpretation along the south profile, paragraph 2).

Figure 10. The diffraction on the shot gather is observed at 0.5 s, while on the stack section is observed at 2.5 km. Can you explain how is diffraction observed so dip, should not be seen around 1.5 – 2 km? Why do you not compare the shot gather with an unmigrated stacked section (in the time domain)?

*Reply to comment:*

We explained in "section 5. Cross-dip analysis" that CDMO stacked sections are considered. Now, Figures 7-11 are updated to show time for depth variations. Table 3 shows that velocity of 6000 m/s is considered for time-to-depth conversion after migration shown in Figures 5-6. The diffraction in Figure 10b is imaged in at ~ 0.75 s which will be ~ 2.5 km.

L.522 ~8000 ms-1

Text has been edited (section 6.2.2 Seismic interpretation along the south profile, paragraph 3)

L.525 of

The sentence is correct and it has not been changed regarding this comment (section 6.2.2 Seismic interpretation along the south profile, paragraph 3): "This uncertainty would suggest greater complexity of the Guercheville fault off the plane of the south profile".

L. 544 remove point before (Mathieu et al., 2020b)

Text has been edited (section 6.2.2 Seismic interpretation along the south profile, paragraph 6)

Figure 12. Why the CDP dash line is so short? What does it means? Also, the change on chs2 is observed more towards the right of the dash line. Is it well located?

*Reply to comment:*

The CDP location is marked based on its coordinate. The dashed line is only presenting the location of CDP without any indication of length or width of the CDP. The projection of Reflection chs2 in shot gather differs compare with its projection and in stacked/migrated sections is different. The shot gather shows the reflections based on the signal gathered in receiver locations. Only after application of NMO corrections, DMO corrections, stacking, and migration process reflections will appear in their true geometrical location.

Section 6.3 can you provided a geological sketch that correlated the north and south interpretations?

*Reply to comment:*

In above we mentioned that the distance between south and north profile is about 50 km. Providing such regional model/interpretation to cover the area from south profile to north profile is beyond the scope of our study. However, interpretation of the regional seismic profile in Chibougamau area including the geological sections or regional models are published somewhere else by some of the co-authors of our paper:

 Mathieu, L., Snyder, D.B., Bedeaux, P., Cheraghi, S., Lafrance, B., Thurston, P., and Sherlock, R.: Deep into the Chibougamau area, Abitibi Subprovince: structure of a Neoarchean crust revealed by seismic reflection profiling, Tectonics, 38, 1–25, 2020.

L.586 missing spaces "… processing work flow applied in this study ..."

Text has been edited (section 7. Conclusions, paragraph 2)

L. 596 (Vermeer, 1990, 1998 and 2010)

Text has been edited (Appendix A, paragraph 1)

References

The following citations are missing in the reference list

Cheraghi et al. (2011); Bellefleur et al. (2018); David et al. (2011); Daigneault and Allard (1990); Bedeaux et al. (2020); Vermeer (1994); Dimroth et al., 1995

*Reply to comment:*

Bedeaux et al. (2020) has ben added to the reference list.

Bellefleur et al. (2018): The correct reference is Bellefleur et al. (2019) which is already in the reference list. Bellefleur et al. (2018) has been changed to Bellefleur et al. (2019) in section 1. Introduction, paragraph 2.

Cheraghi et al. (2011): This reference has been added to the reference list.

Daigneault and Allard (1990): The proper reference is Daigneault et al. (1990) which is already in the reference list and text has been changed to properly cite this reference in section 2. Geological setting, paragraph, 3.

Dimroth et al., 1995: The proper reference is Dimroth et al. (1985) which is already in the reference list. Dimroth et al. (1985) is cited in section 2. Geological setting, paragraph, 3.

David et al. (2011) has been added to the reference list.

Vermeer (1994): This reference has been added to the reference list.

The following reference is not cited in the text

Juhlin, C.: Finite difference elastic wave propagation in 2D heterogeneous transversely isotropic media, Geophysical Prospecting, 43, no.6, 843–858, 1995b

This reference has been removed from the reference list.